# Sebra: Debiasing through Self-Guided Bias Ranking

**Adarsh Kappiyath[1], Abhra Chaudhuri[2], Ajay Jaiswal[3],**
**Ziquan Liu[4], Yunpeng Li[1], Xiatian Zhu[1], Lu Yin[1]\***
[1]University of Surrey, [2]Fujitsu Research of Europe, [3]University of Texas at Austin
[4]Queen Mary University of London

## Abstract

Ranking samples by fine-grained estimates of spuriosity (the degree to which spurious cues are present) has recently been shown to significantly benefit bias mitigation, over the traditional binary biased-*vs*-unbiased partitioning of train sets. However, this spuriosity ranking comes with the requirement of human supervision. In this paper, we propose a debiasing framework based on our novel Self-Guided Bias Ranking (*Sebra*), that mitigates biases (spurious correlations) via an automatic ranking of data points by spuriosity within their respective classes. Sebra leverages a key local symmetry in Empirical Risk Minimization (ERM) training – the ease of learning a sample via ERM inversely correlates with its spuriousity; the fewer spurious correlations a sample exhibits, the harder it is to learn, and vice versa. However, globally across iterations, ERM tends to deviate from this symmetry. Sebra dynamically steers ERM to correct this deviation, facilitating the sequential learning of attributes in increasing order of difficulty, *i.e.*, decreasing order of spuriosity. As a result, the sequence in which Sebra learns samples naturally provides spuriosity rankings. We use the resulting fine-grained bias characterization in a contrastive learning framework to mitigate biases from multiple sources. Extensive experiments show that Sebra consistently outperforms previous state-of-the-art unsupervised debiasing techniques across multiple standard benchmarks, including UrbanCars, BAR, CelebA, MultiNLI, and ImageNet-1K. Code, pre-trained models, and training logs are available at https://kadarsh22.github.io/sebra_iclr25/.

## 1 Introduction

Distribution shifts driven by spurious correlations (*aka* biases or shortcuts) are arguably one of the most studied forms of subpopulation shift (Koh et al., 2021; Yang et al., 2023). Models trained on data that have certain *easy-to-learn* attributes, spuriously correlated with labels, can overly rely on such spurious attributes, resulting in suboptimal performance during deployment (Geirhos et al., 2019). Both supervised (Sagawa et al., 2020; Idrissi et al., 2022) and unsupervised (Nam et al., 2020; Liu et al., 2021; Li et al., 2022; Park et al., 2023) methodologies for making neural networks robust to spurious correlations, a task also known as *debiasing*, have been developed. To get around the expensive human labor involved in acquiring bias labels for training supervised debiasing algorithms, unsupervised methods typically take a two-stage approach: an initial stage for bias identification and a second stage for bias mitigation. Unsupervised bias identification often relies on certain characteristics of spurious attributes, such as their relative ease of learning compared to target attributes (Nam et al., 2020), formation of clusters in feature space (Sohoni et al., 2020), adherence to a low-rank property (Huh et al., 2023), etc. This is followed by the mitigation step via resampling (Idrissi et al., 2022), contrastive learning (Zhang et al., 2022), and pruning (Park et al., 2023), etc.

Existing bias identification methods typically categorize data points into two (Nam et al., 2020; Liu et al., 2021) or more discrete groups (Sohoni et al., 2020; Yang et al., 2024). However, they do not offer insights into how the strength of spurious correlations varies across the identified groups, nor do they account for the variation in the strength of spurious correlations across instances within each

---

*Corresponding author.

group. Recent works, such as Singla & Feizi (2022); Moayeri et al. (2023), address these limitations by ranking data points based on spuriosity—the degree to which common spurious cues are present. However, these spuriosity / bias ranking methods rely on human supervision or auxiliary biased models to identify biased features. Furthermore, they heavily rely on the interpretability of neural features extracted from adversarially trained encoders and the effectiveness of the interpretability techniques employed, which limits their applicability.

To address these limitations, we present Self-Guided Bias Ranking (*Sebra*), a spuriosity ranking algorithm without the need for human intervention. Sebra is based on the observation of a local symmetry in ERM (Empirical Risk Minimization) training – in a given iteration, the hardness of learning a sample is inversely correlated with the amount of spurious features it contains. In other words, the lower the amount of spurious features, the harder a sample is to learn, and vice versa. We call this, the Hardness-Spuriosity Symmetry (Assumption 1), which consequently gives rise to a corresponding conservation law (Theorem 1) relating the hardness of learning a sample to a measure of its spuriosity. This implies that the spuriosity ranking can be derived by looking at the trajectory (through the sample space) of a model that learns attributes sequentially in increasing order of hardness. We empirically confirm the validity of the Hardness-Spuriosity Symmetry assumption in Appendix G.2.

However, when training a neural network on samples with varying levels of spuriosity, globally across iterations, ERM tends to deviate from this trajectory due to (a) reliance on spurious features, since higher spuriosity samples are known to inhibit the learning of those with relatively lower levels of spuriosity Qiu et al. (2024) and (b) non-uniform gradient updates received for samples of different levels of spuriosity due to different values of the task loss (influenced by their levels of spuriosity), leading to non-determinism in the order in which samples are learned. Sebra corrects this deviation by steering the optimization pathway of a neural network by dynamically modulating ERM through a pair of controller variables to follow the conservation law corresponding to the Hardness-Spuriosity Symmetry, while minimizing the interference caused by samples of one spuriosity level on the learning of the other, thereby enabling the network to learn attributes in increasing order of difficulty. Consequently, a readout of the order in which samples are learned along this pathway serves as our predicted spuriosity ranking, requiring no human supervision. By leveraging the fine-grained spuriosity rankings obtained through Sebra and incorporating them into a simple contrastive loss, we outperform previous state-of-the-art unsupervised and supervised debiasing techniques across multiple benchmarks, UrbanCars, BAR, CelebA, MultiNLI, and ImageNet-1K.

To summarize, we: ❶ Introduce a novel self-guided bias ranking framework, Sebra, to dynamically rank the data points of each class on the decreasing order of the strength of spurious signals, without any human supervision; ❷ Utilize these derived rankings to enable debiased learning using a simple contrastive learning framework; ❸ Empirically demonstrate the effectiveness of our proposed approach across multiple datasets with spurious correlations, including UrbanCars, CelebA, BAR, MultiNLI and ImageNet-1K.

## 2 RELATED WORKS

**Bias Identification:** A plethora of methods assume knowledge of bias either in the form of bias labels Lee et al. (2021); Idrissi et al. (2022) or type of bias Geirhos et al. (2019); Chang et al. (2021). Even though these methods produce superior debiasing results, obtaining bias annotations for all biases or identifying the type of bias requires significant human efforts. This led to the development of various inductive biases suitable for bias identification. One of the most commonly used inductive biases for bias identification is the property of bias being easier to learn. In Nam et al. (2020), bias is identified by obtaining a bias-only model through upweighting data points that are easy to learn. Another popular bias identification strategy relies on training a model with a limited network capacity using empirical risk minimization Liu et al. (2021), the hypothesis being that a model with a small capacity would face difficulties in learning complex features and thus prefer to learn easy spurious features. Such simple bias identification schemes has shown to be very useful for datasets with single bias attributes but encounter *Whac-A-Mole dilemma* Li et al. (2023) when faced with datasets with multiple spurious correlations. In Sohoni et al. (2020); Yang et al. (2024), clusters based on biased attributes in the feature space are utilized for bias identification but provide no means to characterize the nature of clusters discovered or how the strength of spurious attributes varies

across these discovered clusters. Recently, Singla & Feizi (2022) proposed a method to identify spurious and core attributes by analyzing neural features of adversarially trained encoders using interpretability techniques like GradCAM and feature attacks. Building on these insights, Moayeri et al. (2023) rank instances within a class based on the presence of these identified attributes, sorting data in decreasing order of spuriosity. Although these methods offer a detailed characterization of spurious attributes in the dataset, their dependence on human supervision and the quality of interpretability techniques used can restrict their applicability. In contrast, our proposed ranking framework orders data in decreasing order of easiness to learn as perceived by an ERM model, without relying on human supervision or fragile interpretability techniques.

**Bias Mitigation:** Some of the simple bias mitigation strategies involve up-weighting bias conflicting points and down-weighting bias aligned points, thereby promoting the model to learn target features from the data Liu et al. (2021); Idrissi et al. (2022); Lee et al. (2021). Other approaches include obtaining a debiased model by training a model to learn different mechanisms to that of a bias-only model Nam et al. (2020), pruning Park et al. (2023) or forgetting the bias information from a biased model Tiwari & Shenoy (2023). In the presence of group labels either inferred or via supervision, a debiased model is obtained by minimizing worst group risk Sagawa et al. (2020). Although simple upweighting methods have shown to be very effective in debiasing they lead to the underutilization of diversity of the training data resulting in suboptimal performance. With the more fine-grained bias identification scheme, we utilise the available data more efficiently using contrastive loss to facilitate debiasing. Contrastive learning effectively debiases data Zhang et al. (2022); Jung et al. (2023), but our ranking scheme refines pair selection, boosting debiasing performance and scalability to diverse and large-scale datasets.

# 3 METHODOLOGY

In this section, we introduce a novel spuriosity-ranking framework, Sebra, designed to rank or order data points in decreasing order of spuriosity. At its core, the framework integrates self-guided weighting mechanisms into the standard Empirical Risk Minimization (ERM) using cross-entropy loss, creating an objective that prioritizes data points by their spuriousness. These self-guided weighting mechanisms guide ERM consistently along a pathway wherein attributes are learned sequentially in the increasing order of hardness. As a result, the order in which instances transition from unlearned to learned naturally reflects the spuriosity of the data point. We demonstrate the effectiveness of this ranked dataset for debiasing within a contrastive learning framework. Our approach is formalized in Section 3.1, with a diagrammatic illustration in Fig. 1.

## 3.1 SEBRA: SELF-GUIDED BIAS RANKING

**Intuition behind Sebra:** Following the example in Fig. 1, consider the problem of classifying "cows" and "camels", where in the train set, cows are spuriously correlated with "green" backgrounds (such as grasslands) in "daylight", and camels are spuriously correlated with the "desert" background at "nighttime". Now, a model trained with ERM tends to classify the training datapoints first based on the background, *i.e.*, cows on grasslands *v.s.* camels on deserts, which implies that it is the easiest attribute to learn Nam et al. (2020). However, when samples exhibiting the background spurious correlation are dropped out from the training set, ERM learns to classify based on the lighting conditions, *i.e.*, cows in daylight *v.s.* camels at nighttime. Finally, it is only when these are also dropped from the training set does the model finally capture the core attributes of cows and camels. Thus, when controlled with an appropriate steering mechanism (dropping of training samples corresponding to the already-learned spurious attribute), the sequence in which ERM learns data points follows a *high-spuriosity to low-spuriosity pathway*, naturally providing a spuriosity ranking. This fine-grained ordering can then be exploited through contrastive learning for debiasing.

**Notations:** Given a train set $X = \{(x_i, y_i)\}_{i=1}^{N}$ with $N$ data points across $C$ classes, we aim to rank them in decreasing order of spuriosity, *i.e.*, if $x_i$ exhibits spurious cues than $x_j$, then $\rho(x_i) < \rho(x_j)$, where $\rho(x)$ is an integer in $[0, N]$ indicating the spuriosity rank of $x$. We use a neural network $f_\theta$ with parameters $\theta$ to drive the ranking process. All proofs and derivations are provided in Appendix A.

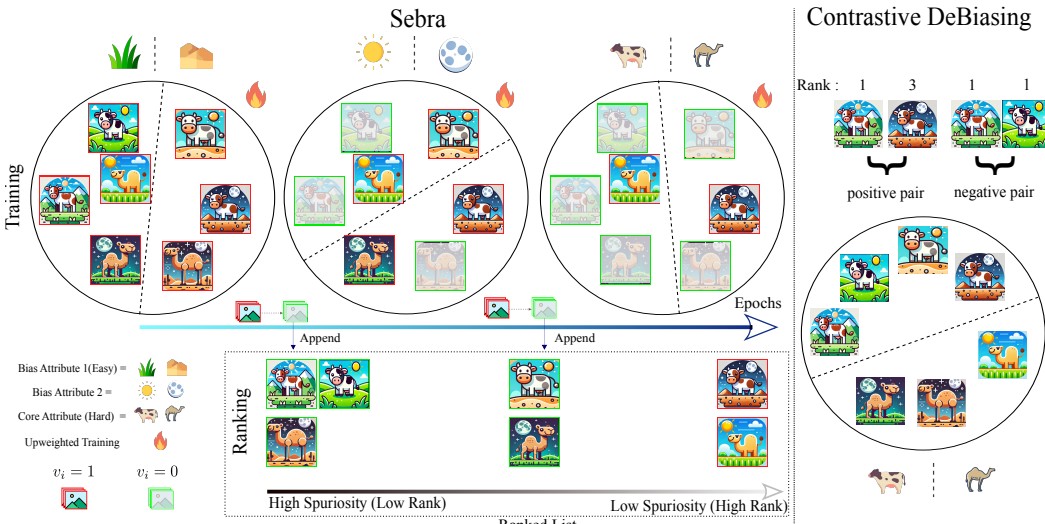

Figure 1: In each step of Self-guided bias ranking (Sebra), datapoints are upweighted with $u_i$ and then trained via ERM. Following this, we estimate $v_i$ for each sample to select them for subsequent training. Samples for which $v_i$ transitions from 1 to 0 are ranked at each step and eliminated from subsequent training. Any unranked samples are appended to the ranked list at the end of the training phase. In the mitigation phase, negative pairs are formed using samples with the same rank, while positive pairs are obtained using samples with a higher rank than the reference samples.

**Definition 1.** For a sample $x \in X$, let $F_x$ be the set of all features types / attributes in $x$. An attribute space $\mathcal{A}$ is the exhaustive collection of all feature types across all $x \in X$, *i.e.*,

$$\mathcal{A} = \bigcup_{x \in X} F_x$$

**Definition 2** (Attribute Types and Spuriosity Ranking). A *causal attribute* $a_c \in \mathcal{A}$ is one that is responsible for determining the label $y$ of a datapoint $x, \forall x \in X$. A *spurious attribute* $a_s \in \mathcal{A}$ is a non-causal attribute that does not determine the label $y$ of any sample $x \in X$, but co-occurs frequently with $a_c$ in $X$. We call the subspace of $\mathcal{A}$ covering all spurious features, $\mathcal{A}_s$, the *spuriosity basis*. The *spuriosity measure* $\mu(x)$ on $X$ is the fraction of spurious attributes in $\mathcal{A}_s$ spanned by the feature-type set $F_x$ of a sample $x \in X$. A *spuriosity ranking* $\rho(x)$ is an ordering on $X$ such that:

$$\rho(x_i) < \rho(x_j) \ \ \forall i, j \leq N \ \mid \ \mu(x_i) > \mu(x_j)$$

In other words, samples with high levels of spuriosity appear earlier in the ranking via $\rho$ than samples with lower levels of spuriosity.

**Assumption 1** (Hardness-Spuriosity Symmetry). The hardness of learning a sample, and its corresponding spuriosity measure, are symmetric to each other – the harder it is to learn a sample, the lower its spuriosity measure, and vice versa.

**Implementation of** $\rho(x)$: We leverage the Hardness-Spuriosity Symmetry to design a form of self-guided bias identification, steering ERM consistently along a *high-spuriosity to low-spuriosity pathway*.

This results in the rank of a data point $x_i$ being the epoch in which its cross-entropy loss (or a monotonically increasing function of it) drops below a certain threshold, or in other words, its predicted probability $p_y$ (or a monotonically increasing function of it) of the correct class $y$ exceeds a certain threshold, determined by hyperparameters, as discussed below.

**Fine-Grained Rank Resolution:** Note that this specific criterion of ranking maps the $N$ datapoint to $M$ buckets, where $M \leq N$. In other words, multiple datapoints can get mapped to the same rank bucket, if they transition below the loss / probability threshold together in the same iteration. However, in our implementation, we also provide information about the spuriosity measure $\mu(x)$ for every data point $x_i$ through a weighting factor called $u_i \propto \mu(x)$. Since $\mu(x)$ is a continuous-valued function, sorting in the decreasing order of $\mu(x)$ provides a straightforward mechanism for

collision resolution and obtaining a fine-grained ranking among data points that inhabit the same coarse-grained rank bucket.

### 3.1.1 FORMULATION

Our ranking algorithm involves the following three key phases in each epoch:

①️ **Selection:** We design the selection mechanism to shift the model's focus to a new subgroup once a particular subgroup has been learned, for it to capture attribute types in order of increasing difficulty across iterations. The training set is partitioned into samples that have been learned, *i.e.*, easier samples with high spuriosity, and those that have not yet been learned, *i.e.*, difficult samples with low spuriosity. The latter are carried forward for further updates to $\theta$ via ERM. This segregation-based selection serves a dual purpose: it mitigates the influence of highly spurious features on the learning of the less spurious ones, and it promotes the learning of attributes in increasing order of difficulty. To implement this, we introduce a binary selection variable $v_i$ for each point $x_i$, which identifies a minimal subset of data points that maximizes the cross-entropy loss:

$$\min_{\theta} \max_{v} \sum_{i=1}^{N} \left\{ v_i^t \mathcal{L}_{\text{CE}}(f_\theta(x_i), y_i) - \lambda v_i^t \right\}$$

$v_i^t \mathcal{L}_{CE}(f_\theta(x_i), y_i)$ is responsible for selecting points that have not yet been learned (*i.e.*, those with a high $\mathcal{L}_{CE}$), while the $-\lambda v_i$ prevents the trivial solution where all $v_i^t$s are set to 1, minimizing the number of points that are selected in a single epoch.

Furthermore, to mitigate the influence of previously learned highly spurious attributes on subsequent learning, we condition the optimization on the state of $v_i$ in the previous iteration, *i.e.*, on $v_i^{t-1}$ as follows:

$$\min_{\theta} \max_{v} \sum_{i=1}^{N} v_i^{t-1} \left\{ v_i^t \mathcal{L}_{\text{CE}}(f_\theta(x_i), y_i) - \lambda v_i^t \right\},$$

and additionally restrict the domain of $v_i^t$ to $\{0, v_i^{t-1}\}$ (instead of the general binary $\{0, 1\}$), where $v_i^0 = 1, \; \forall i \in [1, N]$. This dynamic domain constraint follows from the order on $X$ induced by the measure $\mu$. It effectively implements the inductive bias that points with higher bias would always be learned before points with fewer spurious features, leading to the result that once something has been learned and ranked (with their corresponding $v_i^t$ set to 0), they need not be considered anymore. Note, however, that before $v_i^{(t-1)}$ becomes 0, (*i.e.*, while it is still 1), solving for the optimal $v_i^t$ is still effectively a general binary optimization problem on $\{0, 1\}$.

②️ **Upweighting:** Next, to counteract the non-uniform gradient updates inherent in ERM, and to facilitate the ranking of points with high spuriosity before those with low spuriosity, we utilize the inductive bias that ERM has a lower local risk, in any given iteration, for high spuriosity samples relative to their lower spuriosity counterparts (Assumption 1). We do so by introducing a weighting variable $u_i$ for each point $x_i$ proportional to the value of the spuriosity measure for that point, $\mu(x_i)$ as follows:

$$\min_{\theta, u} \max_{v} \sum_{i=1}^{N} v_i^{t-1} \left\{ v_i^t u_i \mathcal{L}_{\text{CE}}(f_\theta(x_i), y_i) - \lambda v_i^t \right\}$$

This essentially results in the selection of those points with the lowest $\mathcal{L}_{\text{CE}}$ and having them ranked before any of the other points with higher values of $\mathcal{L}_{CE}$ (more difficult-to-learn points, and hence, with fewer spurious features). In principle, following from Assumption 1, $u$ could be any monotonically decreasing function of $\mathcal{L}_{\text{CE}}$.

However, a shortcut solution to minimizing $u$ is to set all $u_i = 0$. We prevent this shortcut by incorporating the inductive bias that $u_i \propto \mu(x_i)$ into the optimization objective. For our specific case, we use $u_i = e^{-t(\mathcal{L}_{\text{CE}}(f_\theta(x_i), y_i))}$, which has the effect that samples with high learnability / spuriosity are upweighted by an exponential function of their spuriosity, where $t(x)$ is a monotonically increasing function of x. We incorporate this constraint into the objective as follows:

$$\min_{\theta, u} \max_{v} \sum_{i=1}^{N} v_i^{t-1} \left\{ v_i^t u_i \mathcal{L}_{\text{CE}}(f_\theta(x_i), y_i) - \lambda v_i^t + \beta g(u_i) \right\}, \tag{1}$$

where $\beta$ is a hyperparameter determining the weight of this constraint, and $g(u_i)$ is a convex function meant to impose the constraint, whose form we uncover next.

**Theorem 1** (Hardness-Spuriosity Conservation). *Iff the spuriosity measure $u_i^* = e^{-t(\mathcal{L}_{CE}(f_\theta(x_i),y_i))}$, where $t(x)$ is a monotonically increasing function of $x$, the variable $u_i$ in Eq. (1), across all values of $\mathcal{L}_{CE}(f_\theta(x_i),y_i)$, satisfies the following conservation law:*

$$u_i\mathcal{L}_{CE}(f_\theta(x_i),y_i) + \beta(u_i\ln u_i - u_i) = c,$$

*such that $u_i^*$ is the minimizer of the conserved function.*

*Intuition:* Theorem 1 arises as a consequence of the Hardness-Spuriosity Symmetry (Assumption 1), which requires the measures of hardness ($\mathcal{L}_{CE}$) and spuriosity ($u_i$) to balance each other out. It states that, for the solution to Eq. (1) to have the form $e^{-t(\mathcal{L}_{CE}(f_\theta(x_i),y_i))}$, the quantity $u_i\mathcal{L}_{CE}(f_\theta(x_i),y_i) + \beta(u_i\ln u_i - u_i)$ should be conserved, *i.e.*, a constant, for all valid choices of $u_i$. The implication is that the optimization on $u$ should be restricted to the space of those values that follow the conservation law. It formalizes the constraint that we need to impose on $u$ in order to avoid the shortcut of setting all $u_i = 0$.

In other words, the solution to $u$ in Eq. (1) is the minimum in the space of all values that satisfy the conservation law. Based on this, we use $g(u_i) = \beta(u_i\ln u_i - u_i)$ in Eq. (1) to enforce the conservation criterion, and obtain our final objective, which we optimize for all three sets of variables $\theta$, $u$, and $v$:

$$\mathcal{L}_{\text{ranking}}(\theta,u,v) = \sum_{i=1}^{N} v_i^{t-1}\left\{v_i^t u_i\mathcal{L}_{\text{CE}}(f_\theta(x_i),y_i) - \lambda v_i^t - \beta u_i + \beta u_i\ln u_i\right\}$$

$$\min_{\theta,u}\max_{v}\mathcal{L}_{\text{ranking}}(\theta,u,v),$$

③ **Ranking:** Finally, samples with high spuriosity, *i.e.*, the ones that have been already learned and dropped out of the training set in the selection phase, are appended to the rank list. Specifically, in every epoch $t$, we select those $x_i$s for which $v_i = 0$ from the selection step, and append them to a rank list $X_{\text{ranked}}$ (which is initially empty) as:

$$X_{\text{ranked}}^t = X_{\text{ranked}}^{t-1} \parallel R,$$

where $\parallel$ is the concatenation operator between two lists, $R$ is an ordered list of data points $x$ such that $x_i < x_j \implies u(x_i) \geq u(x_j);\ \forall x_i,x_j \in R$ and $v^{t-1}(x) = 1, v^t(x) = 0;\ \forall x \in R$. Below we discuss how Sebra progressively orders training samples based on spuriosity by optimizing the variables associated with the above three phases.

## 3.2 OPTIMIZATION

Based on our formulation in Section 3.1.1, Sebra is parameterized by a set of three variables, $\theta, u$, and $v$, respectively corresponding to the Selection, Upweighting, and Ranking phases. Since they are all independent, one can optimize $\mathcal{L}_{\text{ranking}}$ *wrt* each of the variables by keeping the others fixed. In each iteration, we first solve for $v_i^t$ to select the points that have not yet been sufficiently learnt, compute their corresponding $u_i$s, with which we upweight and minimize $\mathcal{L}_{\text{CE}}$ *wrt* $\theta$, (which is non-zero for only those samples that have been selected by $v$ in the beginning of the iteration), and finally, set aside and rank samples whose $v_i$s switched from 1 to 0 in this iteration to avoid interfering with subsequent rankings.

**Selection:** We start by maximizing $\mathcal{L}_{\text{ranking}}(\theta,u,v)$ *wrt* $v$. Note, here, that solving for $v_i^t$ is a discrete optimization problem, since $v_i^t \in \{0, v_i^{t-1}\}$. Let $k = u_i\mathcal{L}_{CE}(f_\theta(x_i),y_i) - \lambda$. This partitions the search space into two halves, *i.e.*, $k \geq 0$ and $k < 0$, as follows:

$$\max_{v}\mathcal{L}_{\text{ranking}}(v \mid \theta,u) = \max_{v}\sum_{i=1}^{N} v_i^{t-1}\left[v_i^t\underbrace{\{u_i\mathcal{L}_{\text{CE}}(f_\theta(x_i),y_i) - \lambda\}}_{k} + \beta u_i\ln u_i\right]$$

The optimal $v_i$ can be obtained in terms of the predicted probability of the correct class $p_y$ as:

$$v_i^{t*} = \begin{cases} 0, & \text{if } p_y > p_{critical}, \\ 1, & \text{otherwise.} \end{cases} \tag{2}$$

Once the $v_i^t$ for a data point $x_i$ has been set to 0, we consider it as learned, and by the design of the optimization objective, it does not influence the subsequent learning of the remaining points.

**Upweighting and Training:** We then solve for the minimization $\mathcal{L}_{\text{ranking}}(\theta, u, v)$ in $u$:

$$\mathcal{L}_{\text{ranking}}(u \mid \theta, v) = \sum_{i=1}^{N} v_i^{t-1} \left\{ v_i^t u_i \mathcal{L}_{\text{CE}}(f_\theta(x_i), y_i, \theta) - \lambda v_i^t - \beta u_i + \beta u_i \ln u_i \right\}$$

Since $\mathcal{L}(u \mid \theta, v)$ is a convex function, it can be minimized by equating its derivative *wrt* $u$ to 0 and solving the resulting equation (when $v_i^t = 1$), which gives us the minimizer of $\mathcal{L}(u \mid \theta, v)$ as:

$$u_i^* = p_y^{\frac{1}{\beta}} \tag{3}$$

We then optimize the parameters of the neural network $\theta$ as follows via regular mini-batch stochastic gradient descent:

$$\min_\theta \mathcal{L}_{\text{ranking}}(\theta \mid u, v) \implies \theta^t = \theta^{t-1} - \nabla_\theta \mathcal{L}_{\text{ranking}} = \theta^{t-1} - u_i \nabla_\theta \mathcal{L}_{\text{CE}}(f_\theta(x_i), y_i) \tag{4}$$

Note how the gradients of the $\mathcal{L}_{\text{CE}}(f_\theta(x_i), y_i)$ *wrt* $\theta$ are upweighted compared to vanilla SGD, by an exponentially decreasing factor of $\mathcal{L}_{\text{CE}}(f_\theta(x_i), y_i)$, *i.e.*, $p_y^{1/\beta}$. This helps the model to converge and rank datapoints with higher spuriosity before it moves on to those with lower levels of spuriosity.

**Ranking:** Finally, we set aside the samples for which $v_i^{(t-1)*} = 1, v_i^{t*} = 0$, and consider them as ranked by appending them to $X_{\text{ranked}}^{t-1}$, in decreasing order of their corresponding $u_i$s, thus allowing for fine-grained rank resolution. We provide the pseudocode for Sebra in Algorithm 1. Based on the ordering obtained, we proceed in the next section, with formulating a contrastive learning based objective for learning a metric space devoid of spurious correlations.

### 3.3 Contrastive Debiasing

The ranking objective introduced in Section 3.1 generates a class-wise ordering of data points in the order of decreasing spuriosity. This fine-grained ranking can be leveraged for efficient debiasing. To demonstrate its effectiveness, we adopt a contrastive loss-based debiasing approach. While contrastive learning has proven effective for debiasing (Zhang et al., 2022), the fine-grained bias characterization offered by Sebra enables the selection of more informative contrastive pairs. This approach surpasses traditional methods, which rely on simpler bias identification mechanisms, such as GCE or partially trained ERM models. Additionally, contrastive learning-based approaches enable the utilization of the entire training dataset, unlike methods such as DFR Kirichenko et al. (2023), which focus on the least spurious examples. Although the fine-grained bias identification generated by sebra could be integrated into other debiasing strategies like DFR, we opt for a contrastive learning framework to showcase the full potential of Sebra.

Given a randomly sampled data point $x_i$ with rank $r$ from class $c$, we sample another instance $x_n^-$ from the same class $c$ and rank $r$ to form a negative pair $(x_i, x_i^-)$. This is motivated by the ranking objective, which assigns the same rank to data points with similar levels of spurious correlations. To form a positive pair, we pair $x_i$ with an instance of higher rank than $r$, as such instances are less likely to share the same spurious features as $x_i$. Using these contrastive pairs, we learn a debiased representation by optimizing the contrastive loss while simultaneously updating the full model via cross-entropy loss. For a classifier $f_\theta$ with encoder $f_{\text{enc}}$, which maps a data point $x$ to its representation $z = f_{\text{enc}}(x)$, the training objective is:

$$\hat{\mathcal{L}}(f_\theta; x, y) = \hat{\mathcal{L}}_{\text{con}}^{\text{sup}}(f_{\text{enc}}; x, y) + \gamma \hat{\mathcal{L}}_{\text{CE}}(f_\theta(x, y), \tag{5}$$

where $\gamma$ is a weighting coefficient. The supervised contrastive loss with $M$ positive pairs and $N$ negative pairs is:

$$\mathcal{L}_{\text{con}}^{\text{sup}}(x; f_{\text{enc}}) = \mathbb{E}\left[ -\log \frac{\exp(z^\top z_m^+/\tau)}{\sum_{m=1}^{M} \exp(z^\top z_m^+/\tau) + \sum_{n=1}^{N} \exp(z^\top z_n^-/\tau)} \right],$$

where $\tau$ is the temperature coefficent, $z_m^+$, $z_n^-$ and $z^\top$ are the embeddings of positive, negative, and reference samples respectively.

## 4 EXPERIMENTAL SETUP

This section outlines the experimental framework for evaluating the effectiveness of the proposed spuriosity ranking and debiasing approach. We outline the datasets, evaluation metrics, and baselines used, with implementation details and hyperparameters in Appendix J.

**Datasets:** We evaluate the proposed ranking and debiasing strategy on one synthetic and four natural datasets with spurious correlations. The synthetic dataset, UrbanCars Li et al. (2023), focuses on car-type classification with spurious correlations involving the background and co-occurring objects. For natural datasets, we use CelebA Liu et al. (2015), which addresses spurious features like age and gender in predicting emotions (smiling/sad) as per Hu et al. (2023). These datasets contain multiple spurious correlations and bias annotations, enabling the definition of a ground truth rank order of spuriosity and evaluating our method's effectiveness. Additionally, we test on two natural vision datasets, BAR (Nam et al., 2020) and ImageNet-1K Deng et al. (2009), and the natural language dataset MultiNLI to demonstrate scalability across domains. Sample images and dataset details are in Appendix B.

**Evaluation Metrics:** Quantitatively comparing spuriosity rankings is challenging due to the absence of ground truth rankings. For datasets with bias annotations, such as Urban Cars and CelebA, where two biases are present (with Bias A being stronger than Bias B), we define the ground truth ordering as follows: (Bias A aligned, Bias B aligned), (Bias A aligned, Bias B conflicting), (Bias A conflicting, Bias B aligned), (Bias A conflicting, Bias B conflicting). We compute Kendall's tau correlation Kendall (1938) to compare our ranking with this ground truth.

For datasets without bias annotations (e.g., BAR), we propose *Performance Disparity (PD)*:

$$PD = \text{Accuracy}_{\text{Bottom } k}(\mathcal{D}_{\text{test}}) - \text{Accuracy}_{\text{Top } k}(\mathcal{D}_{\text{test}}) \tag{6}$$

PD measures ranking quality by assessing accuracy differences between models trained on highly vs. minimally spurious samples. High PD indicates effective separation of spuriosity levels.

To evaluate debiasing, we adopt and extend three *UrbanCars* metrics Li et al. (2023): **BG Gap, CoObj Gap, and BG + CoObj Gap**, which quantify accuracy drops when spurious features are unaligned. These metrics generalize to other datasets, e.g., **Age Gap, Gender Gap, and Age + Gender Gap** for CelebA. **Avg. GAP** aggregates robustness across shifts. For BAR, we report test accuracy on bias-conflicting samples, following Li et al. (2022). Further metric details are in Appendix E.

**Baselines:** We compare the performance of the proposed approach with a supervised approach Group DRO (Sagawa et al., 2020) and five popular unsupervised approaches ERM (Vapnik, 1999), LfF (Nam et al., 2020), JTT (Liu et al., 2021), Debian (Li et al., 2022) and DFR (Kirichenko et al., 2023). The supervised approaches assume the availability of shortcut labels for all the spurious attributes while the unsupervised methods have access only to target labels. Both classes of methods further assume access to a small supervised validation set for hyperparameter tuning.

### 4.1 RESULTS

In this section, we compare the performance of the proposed method with various baselines and datasets described in Section 4 to demonstrate its effectiveness in spuriosity ranking and debiasing.

**Ranking Evaluation.** As observed in Table 1, the proposed method produces a superior ordering of data points as indicated by a higher value of Kendall's tau-b coefficient for both datasets. The inferior performance of Spuriosity Ranking (Moayeri et al., 2023) despite human supervision could be attributed to the fact that biased attributes like background encompass multiple sub-attributes like lighting, sky, terrain, etc, and different sub-attributes are captured by different neurons rather than one or few neurons. Since these concepts are distributed across multiple neurons they need not be contained in top-k activations Fig. 4, resulting in many of these attributes being not considered while sorting the data.

**Debiasing Evaluation.** As shown in Table 2, our proposed Sebra outperforms all the unsupervised methods in simultaneously mitigating multiple biases, as evidenced by the lower average gap (Avg. GAP) metric across datasets. While Sebra may not always achieve the best performance on

individual Bias GAP metrics, this is due to the *whac-a-mole* dilemma observed in previous methods, where mitigating one bias attribute exceptionally well can amplify the other bias attribute.

This can result in a very low Bias GAP for one attribute, even though the model remains highly biased overall. Furthermore, the proposed method consistently surpasses previous approaches. Additionally, our method performs comparably to prior single-bias unsupervised methods in single-bias settings, highlighting its effectiveness. An extended version of Table 2 is provided in Appendix D.2.

Table 1: Quantitative comparison of Sebra with various baselines. The results are shown in terms of Kendall's $\tau$ for Urban Cars and CelebA, and Performance Disparity (PD) for BAR.

| Method | Urban Cars | CelebA | BAR |
|--------|-----------|--------|-----|
| Metric | *Kendall's $\tau$ ($\uparrow$)* | *Kendall's $\tau$ ($\uparrow$)* | *PD ($\uparrow$)* |
| Random Ordering | 0.02 | -0.01 | 0.25 |
| ERM-based Ranking | 0.12 | 0.14 | 4.55 |
| Spuriosity Ranking | 0.40 | 0.38 | 28.88 |
| Sebra (Ours) | **0.85** | **0.69** | **32.32** |

Table 2: Performance comparison across UrbanCars, CelebA, BAR, ImageNet, and MultiNLI datasets. **Sup.**: Whether the model requires group or spurious attribute annotations (✗: not required, ✓: required). I.D. Acc. measures performance without subpopulation shift, while Avg GAP does so in its presence. All results are reported as mean (standard deviation).

| Methods | Sup. | UrbanCars | | | CelebA | | | BAR |
|---------|------|-----------|--|--|--------|--|--|-----|
| | | I.D. Acc. ($\uparrow$) | WG Acc. ($\uparrow$) | Avg GAP ($\uparrow$) | I.D. Acc. ($\uparrow$) | WG Acc. ($\uparrow$) | Avg GAP ($\uparrow$) | Test Acc. ($\uparrow$) |
| Group DRO | ✓ | 91.60 (1.23) | 75.70 (1.79) | -10.30 (1.35) | 90.08 (0.70) | 37.9 (1.6) | -5.79 (1.63) | - |
| ERM | ✗ | 97.60 (0.86) | 33.20 (0.86) | -31.90 (3.92) | 96.43 (0.13) | 36.0 (1.7) | -22.83 (0.84) | 68.00 (0.43) |
| LfF | ✗ | 97.20 (2.40) | 35.60 (2.40) | -31.06 (3.56) | 95.12 (0.35) | 35.5 (2.0) | -22.57 (1.26) | 68.30 (0.97) |
| JTT | ✗ | 95.80 (1.45) | 33.30 (6.90) | -20.50 (2.61) | 91.86 (1.48) | 38.7 (2.4) | -26.81 (2.53) | 68.14 (0.28) |
| Debian | ✗ | 98.00 (0.89) | 30.10 (0.89) | -31.40 (1.44) | 96.28 (0.37) | 41.1 (4.3) | -22.56 (0.54) | 69.88 (2.92) |
| DFR | ✗ | 89.70 (1.21) | - | -20.93 (2.61) | 60.12 (1.28) | - | -19.16 (3.27) | 69.22 (1.25) |
| Sebra (Ours) | ✗ | 92.54 (2.10) | **73.8 (3.28)** | **-10.57 (1.72)** | 88.61 (3.36) | **65.3 (4.1)** | **-9.82 (3.06)** | **75.36 (2.23)** |

| Method | Sup. | ImageNet-1K | | | | | MultiNLI |
|--------|------|-------------|--|--|--|--|----------|
| | | I.D. Acc. ($\uparrow$) | IN-W Gap ($\uparrow$) | IN-9 Gap ($\uparrow$) | IN-R Gap ($\uparrow$) | Carton Gap ($\uparrow$) | WG. Acc ($\uparrow$) |
| LLE | ✓ | 76.25 | -6.18 | -3.82 | -54.89 | +10 | - |
| ERM | ✗ | 76.13 | -26.64 | -5.53 | -55.96 | +40 | 66.8 |
| LfF | ✗ | 70.26 | -17.57 | -8.10 | -56.54 | +40 | 63.6 |
| JTT | ✗ | 75.64 | -15.74 | -6.75 | -55.70 | +32 | 69.1 |
| Debian | ✗ | 74.05 | -20.00 | -7.29 | -56.70 | +30 | - |
| Sebra (Ours) | ✗ | 74.89 | **-14.77** | **-3.15** | **-54.81** | **+25** | **72.3** |

## 4.2 ANALYSIS AND ABLATION STUDIE

In this section, we present a comprehensive set of analyses and ablation studies to provide deeper insights into the performance of Sebra. Specifically, we investigate how the training dynamics of a model optimized using the proposed ranking objective differ from those of a standard empirical risk minimization (ERM)–based model. This comparison elucidates how the proposed selection and weighting mechanisms modulate the ERM training dynamics to facilitate spuriosity ranking. Furthermore, we conduct ablations on the various components of our framework to quantify their contributions to the overall ranking quality. Additional ablation studies are provided in Appendix G.

**Analysis of Ranking Dynamics:** In Section 3, we introduced Sebra, which integrates targeted modifications to ERM to systematically rank data points in the decreasing order of spuriosity. To rigorously assess the impact of these modifications, we conduct a detailed analysis of the training dynamics under the Sebra objective compared to standard ERM. Specifically, we leverage the UrbanCars dataset, which includes bias annotations, enabling a detailed evaluation of how spurious and intrinsic features are differentially learned across the two training paradigms. In Fig. 2, we plot the accuracy of three visual cues—object (e.g., car body type), background, and co-occurring objects—on the unbiased validation set by comparing the model's {urban, country} predictions to the corresponding labels. As shown in Fig. 2 (Left), both models initially prioritize the easiest bias attribute (background). However, as training progresses, the sebra objective induces a more pronounced forgetting of learned attributes compared to ERM, likely due to the selection mechanism

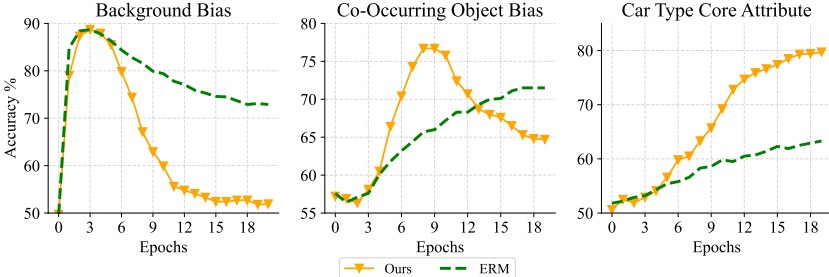

Figure 2: Training dynamics of Sebra and ERM monitored in terms of accuracies of background bias (left), co-occurring object bias (center), and core attribute (right).

through $v_i$ that refocuses the model's attention on other subgroups. The aggressive forgetting under the sebra framework overcomes the slowdown in the convergence of difficult attributes in the presence of simpler correlated features (Qiu et al., 2024), as evidenced by the higher peaks for both target and co-occurring object attributes. Another interesting observation is that, for relatively difficult bias attributes, such as co-occurring objects, the naive ERM formulation struggles to differentiate them from core attributes, as indicated by a simultaneous increase in accuracies in Fig. 2 (center and right). Sebra effectively addresses this challenge by leveraging the upweighting factor $u_i$, which amplifies the influence of highly spurious instances, thereby facilitating the progressive learning of these attributes. Additionally, the self-guided mechanism driven by $u_i$ enhances overall performance, as demonstrated by the ranking improvements shown in Table 3. The non-overlapping peaks indicate that instances with a higher prevalence of respective attributes are assigned different ranks. Therefore, the ranking objective of sebra leads to well-segregated sequential learning of different shortcuts in the decreasing order of spuriosity. This analysis confirms our intuition and provides empirical evidence that the sebra efficiently ranks data in decreasing order of spuriosity.

**Effect of Loss Components:** To evaluate the contribution of various loss components, we conduct an ablation study by systematically removing components and measuring their impact on the quality of the resulting rankings using Kendall's $\tau$ coefficient. The results of this analysis are shown in Table 3. When using only $\mathcal{L}_{CE}$, corresponding to standard ERM training, we observe that ERM cannot alone

Table 3: Ablation study of different components used in Sebra.

| $\mathcal{L}_{\text{CE}}(\theta)$ | $\mathcal{L}_{\text{ranking}}(v_i)$ | $\mathcal{L}_{\text{ranking}}(u_i)$ | Kendall's $\tau$ ($\uparrow$) |
|:---:|:---:|:---:|:---:|
| ✓ | - | - | 0.12 |
| ✓ | ✓ | - | 0.79 |
| ✓ | ✓ | ✓ | **0.85** (Sebra) |

rank data points effectively. To address this, we define a proxy ranking based on the epoch at which the predicted probability of the target attribute surpasses a fixed threshold. As shown in Table 3, the model trained with naive cross-entropy loss exhibits a low correlation with the ground truth ranking, as indicated by the low Kendall's $\tau$. This suggests that naive cross-entropy fails to capture the underlying spuriosity of the data. The slightly positive value of $\tau$ likely reflects ERM's inherent, albeit weak, capacity to asynchronously learn different attributes. When $v_i$ is added to the objective function, the ranking quality improves significantly. This suggests that ERM's poor bias ranking performance may be due to interference from the easiest attributes when learning more complex ones. By incorporating both $v_i$ and $u_i$ into the training objective, the ranking quality further improves to 0.85, underscoring their importance in enhancing performance.

## 5 CONCLUSION AND FUTURE WORKS

We propose a novel debiasing strategy, *Sebra*, based on a fine-grained ranking of data points in decreasing order of spuriosity, obtained without any human supervision. Sebra facilitates spuriosity ranking by modulating the training dynamics of a simple ERM model to iteratively focus on highly spurious data points while simultaneously excluding already ranked datapoints from the ranking process. We further demonstrate how this fine-grained bias ordering enhances bias mitigation, by considering a contrastive learning-based approach as an exemplar on various datasets. Future work could explore bias mitigation strategies tailored to Sebra's rankings, refine the ranking scheme, and develop unsupervised metrics for evaluating spuriosity rankings.

## 6 ACKNOWLEDGMENTS

Adarsh Kappiyath thanks Silpa Vadakkeeveetil Sreelatha for invaluable conversations and feedback throughout this project. Additionally, we would like to thank the University of Surrey for providing the valuable computing infrastructure needed for this project.

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

# A  PROOFS AND DERIVATIONS

## A.1  PROOF OF THEOREM 1

*Proof.* We start by first proving the case when $u_i^* = e^{-t(\mathcal{L}_{CE}(x_i, y_i, \theta))}$ leads to the conservation law, *i.e.*, $u_i^* = e^{-t(\mathcal{L}_{CE}(x_i, y_i, \theta))} \implies u_i \mathcal{L}_{CE}(x_i, y_i, \theta) + \beta(u_i \ln u_i - u_i) = c$.

$$\min_u u_i \mathcal{L}_{CE}(x_i, y_i, \theta) - \lambda v_i^t + \beta g(u_i) \implies \mathcal{L}_{CE}(x_i, y_i, \theta) + \beta g'(u_i) = 0$$

$$\implies g'(u_i) = -\frac{1}{\beta}\mathcal{L}_{CE}(x_i, y_i, \theta) \implies u_i^* = (g')^{-1}(-\frac{1}{\beta}\mathcal{L}_{CE}(x_i, y_i, \theta))$$

Now, we know that $u_i^* = e^{-t(\mathcal{L}_{CE}(x_i, y_i, \theta))}$. Considering $t = \frac{1}{\beta}x$, we have, $(g')^{-1}(x) = e^x \implies g'(x) = \ln x$. Then,

$$\ln u_i^* = -\frac{1}{\beta}\mathcal{L}_{CE}(x_i, y_i, \theta) \implies \int \ln u_i^* \, du_i = -\frac{1}{\beta}\int \mathcal{L}_{CE}(x_i, y_i, \theta) \, du_i$$

$$\implies u_i \mathcal{L}_{CE}(x_i, y_i, \theta) + \beta(u_i \ln u_i - u_i) = c = \lambda v_i^t,$$

since $\lambda v_i^t$ was the constant that vanished under the derivative. This proves the statement in the direction $u_i^* = e^{-t(\mathcal{L}_{CE}(x_i, y_i, \theta))} \implies u_i \mathcal{L}_{CE}(x_i, y_i, \theta) + \beta(u_i \ln u_i - u_i) = c$.

Next, we prove the statement in the other direction, *i.e.*, when minimizing the conserved function leads to the exponentially decreasing characteristic of $u_i^*$. Solving for the minimizer of the conservation expression, we get:

$$u_i^* = \min_u u_i \mathcal{L}_{CE}(x_i, y_i, \theta) + \beta(u_i \ln u_i - u_i) - \lambda v_i^t \implies \mathcal{L}_{CE}(x_i, y_i, \theta) + \beta \ln u_i = 0$$

$$\implies \ln u_i^* = -\frac{1}{\beta}\mathcal{L}_{CE}(x_i, y_i, \theta) \implies u_i^* = e^{-\frac{1}{\beta}\mathcal{L}_{CE}(x_i, y_i, \theta)} = e^{-t(\mathcal{L}_{CE}(x_i, y_i, \theta))},$$

where $t = \frac{1}{\beta}x$. This proves the statement in the direction $u_i \mathcal{L}_{CE}(x_i, y_i, \theta) + \beta(u_i \ln u_i - u_i) = c \implies u_i^* = e^{-t(\mathcal{L}_{CE}(x_i, y_i, \theta))}$, and completes the proof of the theorem. $\square$

## A.2  SOLUTION FOR $u_i$

$$\frac{\partial}{\partial u_i}\mathcal{L}_{ranking}(u \mid \theta, v) = 0 \implies v_i^t \mathcal{L}_{CE}(x_i, y_i, \theta) - \beta + \beta[1 + \ln u_i] = 0$$

$$\implies \ln p_y = \beta \ln u_i \implies \ln u_i = \frac{1}{\beta}\ln p_y \implies \ln u_i = \ln p_y^{1/\beta} \implies u_i^* = p_y^{1/\beta}$$

## A.3  SOLUTION FOR $v_i$

**Case 1** ($k \geq 0$)**:** When $k \geq 0$, the optimal solution is $v_i^{t*} = 1$, ensuring $\mathcal{L}(v \mid \theta, u) \geq 0$. Otherwise, $\mathcal{L}_{ranking}(v \mid \theta, u) = 0$, which is always less than or equal to when $v_i^t = 1$. Thus, $v_i^t = 1$ is the maximizer of $\mathcal{L}_{ranking}(v \mid \theta, u)$ when $k \geq 0$.

Below, we derive the condition for optimality in terms of the predicted probability of the correct class $p_y$ (this applies only when $v_i^{t-1} = 1$):

$$u_i \mathcal{L}_{CE}(x_i, y_i, \theta) - \lambda \geq 0 \implies u_i \ln p_y \leq -\lambda \implies p_y \leq e^{-\lambda/u_i}$$

**Case 2** ($k < 0$)**:** When $k < 0$, the optimal solution is $v_i^{t*} = 0$. If $v_i^t = 0$, $\mathcal{L}_{ranking}(v \mid \theta, u) = v_i^t k$ would be negative. Thus, $v_i^t = 0$ maximizes $\mathcal{L}_{ranking}(v \mid \theta, u)$ in this case. Similarly to Case 1, we derive the condition for optimality when $k < 0$ in terms of the predicted probability of the correct class $p_y$:

$$p_y > e^{-\lambda/u_i}.$$

We consider the inequality $p_y > e^{-\lambda/p_y^{1/\beta}}$, where $\lambda$ and $\beta$ are constants, and aim to rewrite the relationship in an analytically solvable form. Taking the natural logarithm of both sides of the inequality, we obtain

$$\ln(p_y) > -\frac{\lambda}{p_y^{1/\beta}}. \tag{7}$$

By multiplying through by $p_y^{1/\beta}$ (valid for $p_y > 0$), this simplifies to

$$p_y^{1/\beta} \ln(p_y) > -\lambda. \tag{8}$$

To simplify further, we introduce the substitution $z = p_y^{1/\beta}$, which implies $p_y = z^\beta$. Substituting into the inequality gives

$$z \ln(z) > -\frac{\lambda}{\beta}. \tag{9}$$

At this point, we solve for $z$ explicitly. The equation $z \ln(z) = C$ (where $C = -\frac{\lambda}{\beta}$) is a well-known form that can be solved using the Lambert $W$-function. The Lambert $W$-function is defined as the inverse of $ye^y = x$, such that $y = W(x)$. Rewriting $z \ln(z)$ in exponential form, we obtain

$$\ln(z) = W\left(-\frac{\lambda}{\beta}\right). \tag{10}$$

Exponentiating both sides gives the explicit solution for $z$:

$$z = e^{W\left(-\frac{\lambda}{\beta}\right)}. \tag{11}$$

Returning to the original substitution $z = p_y^{1/\beta}$, we substitute back to express $p_y$ in terms of the Lambert $W$-function:

$$p_y^{1/\beta} = e^{W\left(-\frac{\lambda}{\beta}\right)}. \tag{12}$$

Finally, raising both sides to the power of $\beta$ gives the solution for $p_y$:

$$p_y = \left(e^{W\left(-\frac{\lambda}{\beta}\right)}\right)^\beta. \tag{13}$$

This expression represents the critical value of $p_y$ where the equality $p_y = e^{-\lambda/p_y^{1/\beta}}$ holds. Thus, the inequality $p_y > e^{-\lambda/p_y^{1/\beta}}$ can now be interpreted as

$$p_y > \left(e^{W\left(-\frac{\lambda}{\beta}\right)}\right)^\beta. \tag{14}$$

The Lambert $W$-function provides an explicit solution for equations where a variable appears both inside and outside of a logarithmic or exponential function. This result establishes the threshold $p_{\text{critical}}$, defined as

$$p_{\text{critical}} = \left(e^{W\left(-\frac{\lambda}{\beta}\right)}\right)^\beta, \tag{15}$$

which represents the critical value of $p_y$ where equality holds. Due to the monotonically decreasing nature of exponential, the inequality

$$p_y > e^{-\frac{\lambda}{p_y^{1/\beta}}}$$

can thus be rewritten as:

$$p_y > p_{\text{critical}}, \tag{16}$$

where $p_{\text{critical}}$ depends only on the constants $\lambda$ and $\beta$ through the Lambert $W$-function. This reformulation establishes that for the inequality $p_y > e^{-\lambda/p_y^{1/\beta}}$ to hold, $p_y$ must strictly exceed the threshold $p_{\text{critical}}$.

## B    Datasets

We evaluate the proposed method across three distinct datasets, each designed to explore different facets of bias and debiasing techniques. Below, we provide a succinct overview of each dataset:

1. **UrbanCars** Li et al. (2023): This synthetic dataset is purposefully crafted to investigate debiasing methodologies amidst multiple spurious correlations. Comprising two classes - UrbanCar and Country Car - each class encompasses 4000 samples. The dataset is characterized by two biased attributes: Background and Co-Occurring Object. UrbanCars feature city-like backgrounds with co-occurring objects such as traffic signs and fire hydrants, while Country Cars are set against rural backgrounds, predominantly featuring animals. UrbanCars is publicly available on Kaggle.

2. **CelebA**: A versatile dataset featuring celebrity faces alongside 40 binary attributes. We focus on the 'smile' attribute as the target, with biases introduced by age and gender. This configuration was introduced in Hu et al. (2023), and we employ their open-source code to obtain the data.

3. **Biased Action Recognition (BAR)** Nam et al. (2020): The Biased Action Recognition (BAR) dataset contains real-world images categorized into six action classes, each biased towards particular locations. The dataset includes six prevalent action-location pairs: Climbing on a Rock Wall, Diving underwater, Fishing on a Water Surface, Racing on a Paved Track, Throwing on a Playing Field, and Vaulting into the Sky. The testing set is composed exclusively of samples with conflicting biases. Therefore, achieving higher accuracy on this set signifies improved debiasing performance.

4. **ImageNet-1K** Deng et al. (2009): ImageNet-1K is a large-scale dataset consisting of over one million high-resolution images categorized into 1,000 distinct object classes. The dataset includes a diverse range of real-world objects across various categories such as animals, vehicles, and everyday items, making it a benchmark for visual recognition tasks. ImageNet-1K is widely used for training and evaluating deep learning models, and achieving high accuracy on this dataset demonstrates strong generalization and recognition capabilities, with an emphasis on overcoming challenges posed by large-scale, varied data.

5. **MultiNLI** Williams et al. (2018): The MultiNLI (Multi-Genre Natural Language Inference) dataset is a large-scale benchmark for evaluating models on the task of natural language inference (NLI). It consists of 433k sentence pairs drawn from a wide variety of genres, including government, fiction, and telephone conversations, among others. Each pair is annotated with one of three labels: entailment, contradiction, or neutral. MultiNLI is designed to test a model's ability to generalize across diverse linguistic contexts, and achieving high performance on this dataset indicates a model's robustness in understanding complex sentence relationships and overcoming genre-specific biases in natural language processing tasks.

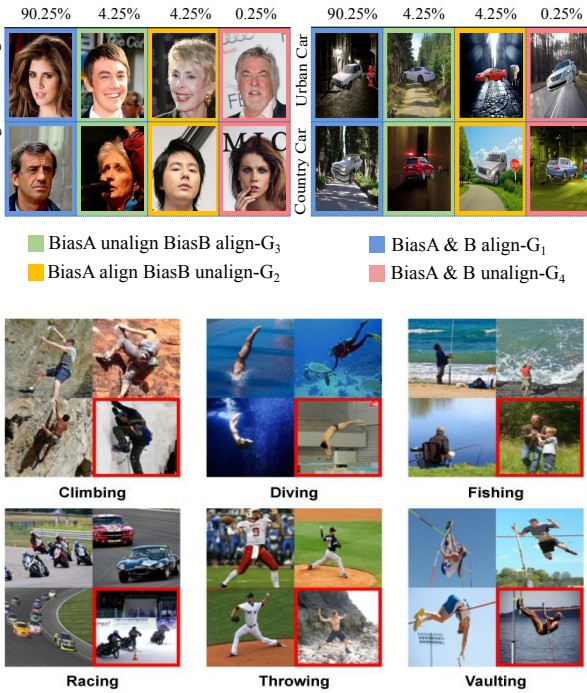

Figure 3: **Dataset samples:** Images from various datasets with multiple spurious correlations used in our experiments are shown below. For CelebA and UrbanCars dataset each column depicts multiple groups categorised based on biased features, as well as their proportions in the training set, each row displays samples from various classes. Images at the bottom demonstrates samples from BAR dataset from 6 classes. The images with red border lines belong to BAR evaluation set, and others belong to BAR training set.

## C  BASELINES

We evaluate the proposed method against a series of unsupervised and supervised bias mitigation techniques. Below, we provide a concise overview of each method:

1. **GroupDRO:** Sagawa et al. (2020) A supervised bias mitigation technique leveraging group labels to identify and mitigate biases across various groups in the training data. The objective is to minimize the worst group accuracy across the identified groups.

2. **ERM** Vapnik (1999): Empirical Risk Minimization, employing cross-entropy loss and $l_2$ regularization.

3. **Learning from Failure (LfF):** Nam et al. (2020) This approach utilizes the Generalized Cross-Entropy (GCE) loss to derive a bias-only model. Subsequently, it learns a debiased model by reweighting the bias-conflicting points to learn a debiased model.

4. **JTT:** Liu et al. (2021) This method uses a ERM model trained for few epochs and identifies the misclassifications obtained by the model as bias conflicting samples and is upweighted for debiased learning.

5. **Debian:** Li et al. (2022) Introducing a novel bias identification scheme relying on the equal opportunity violation criteria, followed by bias mitigation strategies.

6. **DFR**: (Kirichenko et al., 2023) demonstrates that ERM model captures non-spurious attributes even when trained with biased training data and thus simple last layer retraining with unbiased data is sufficient for debiasing.

# D RESULTS

## D.1 QUALITATIVE RESULTS

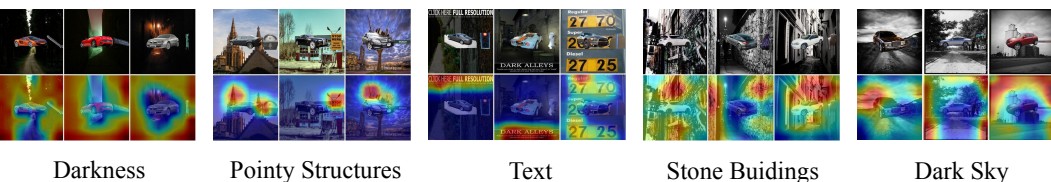

Figure 4: Top 5 spurious concepts discovered using Spuriosity rankings introduced in Moayeri et al. (2023). As observed, the identified neurons capture only a subset of features corresponding to the spurious attribute 'background'; thus, ranking relying on top-k highly activating neurons would only rely on partial characteristics of spurious features.

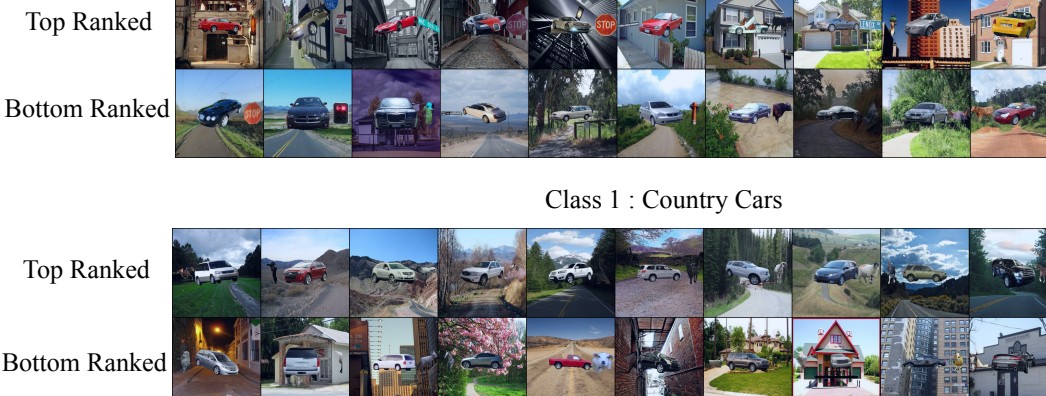

Figure 5: Qualitative Analysis on UrbanCars Dataset: Examples of top-ranked (high-spurious) and bottom-ranked(low-ranked) samples as ranked by Sebra, showcasing a range of samples from both the classes.

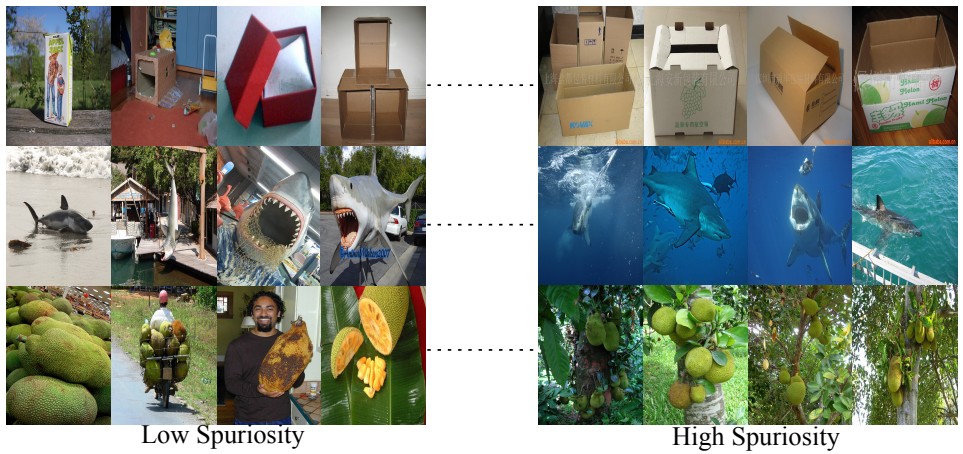

Figure 6: **Qualitative Analysis on the ImageNet-1K Dataset.** Sebra identifies spurious correlations in three classes. In the 'Carton' class, Sebra detects watermark shortcuts, while in the 'Shark' and 'Jackfruit' classes, deep-sea and tree backgrounds are flagged as spurious features.

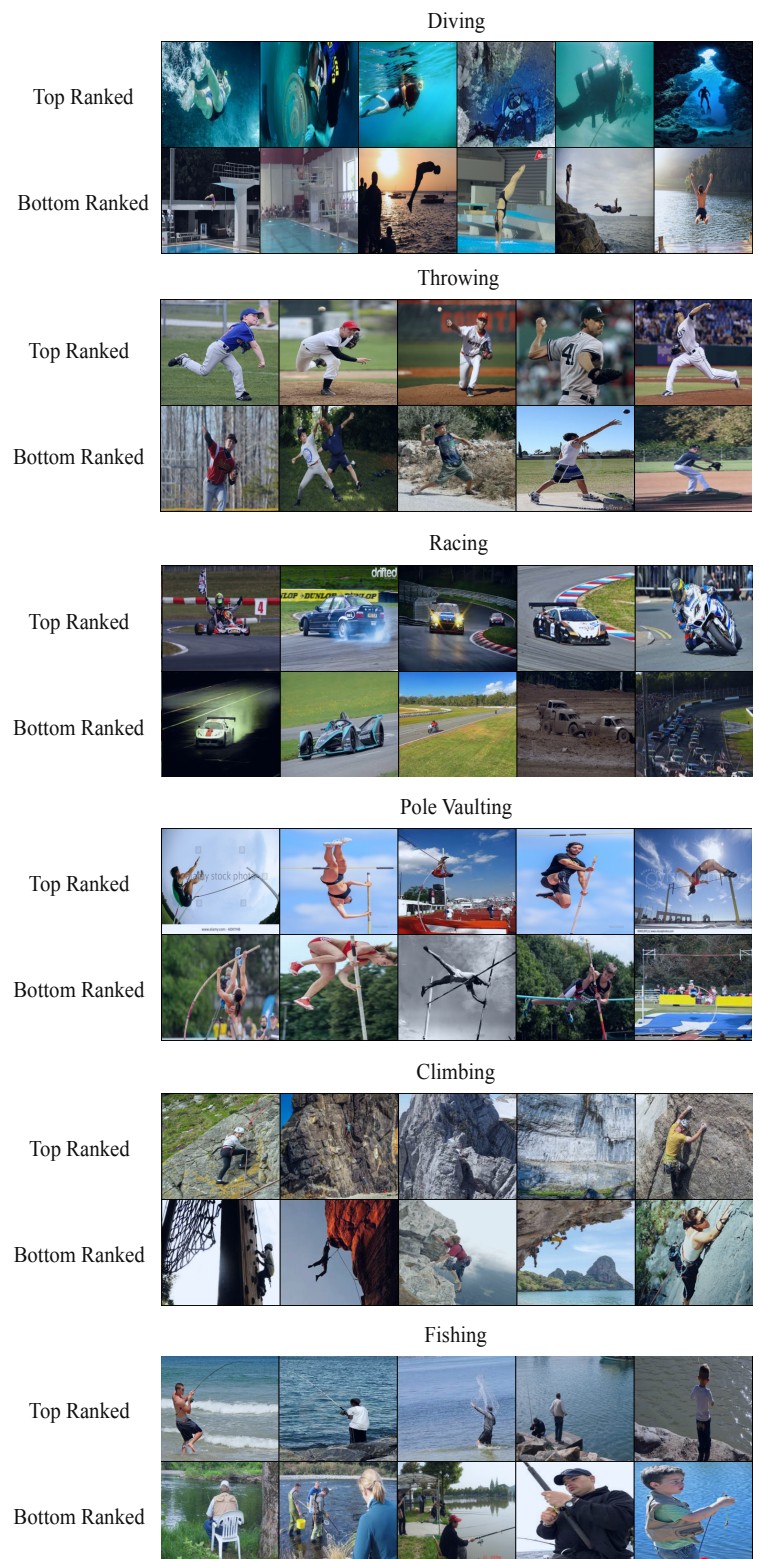

Figure 7: Qualitative Analysis on BAR Dataset: Examples of top-ranked and bottom-ranked samples as ranked by Sebra, showcasing a range of samples across different classes.

## D.2 QUANTITATIVE RESULTS

Table 4: Performance comparison on the UrbanCars dataset. Sup.: Whether the model requires group or spurious attribute annotations in advance (✗: not required, ✓: required). The best-performing results among unsupervised methods are marked in bold.The baseline results are taken from Li et al. (2023)

| Methods | Sup. | I.D. Acc. (↑) | BG GAP (↑) | CoObj GAP (↑) | BG + CoObj GAP (↑) |
|---------|------|---------------|------------|----------------|---------------------|
| Group DRO | ✓ | 91.60(1.23) | -10.90 (1.08) | -3.60 (0.19) | -16.40 (2.80) |
| ERM | ✗ | 97.60 (0.86) | -15.30 (1.35) | -11.20 (5.07) | -69.20 (5.34) |
| LfF | ✗ | 97.20 (2.40) | -11.60 (1.23) | -18.40 (4.01) | -63.20 (2.21) |
| JTT | ✗ | 95.80 (1.45) | -8.10 (1.08) | -13.30 (4.28) | -40.10 (2.48) |
| Debian | ✗ | 98.00 (0.89) | -14.90 (1.08) | -10.50 (1.47) | -69.00 (1.78) |
| DFR | ✗ | 89.70 (1.21) | -10.70 (1.85) | -6.90 (2.56) | -45.20 (3.42) |
| Sebra | ✗ | 92.54 (2.10) | **-6.54 (1.38)** | -7.84 (1.38) | **-17.34 (2.40)** |

Table 5: Performance comparison on the CelebA and BAR. Sup. indicates whether the method is supervised for bias (✓) or not (✗). The best results among unsupervised methods are marked in bold.

| Methods | Sup. | CelebA | | | | BAR |
|---------|------|--------|--|--|--|-----|
| | | I.D. Acc (↑) | Gender GAP (↑) | Age GAP (↑) | Gender+Age GAP (↑) | Test Acc. (↑) |
| Group DRO | ✓ | 90.08 (0.70) | -5.67 (2.23) | -2.6 (2.4) | -9.11 (3.34) | - |
| ERM | ✗ | 96.43 (0.13) | -22.7 (1.34) | -2.03 (0.77) | -43.77 (0.42) | 68.00 (0.43) |
| LfF | ✗ | 95.12 (0.35) | -24.14 (1.28) | -1.33 (1.2) | -42.26 (1.32) | 68.30 (0.97) |
| JTT | ✗ | 91.86 (1.48) | -31.07 (1.21) | -3.51 (2.44) | -45.85 (3.93) | 68.14 (0.28) |
| Debian | ✗ | 96.28 (0.37) | -22.03 (1.26) | -3.23 (1.65) | -42.41 (0.49) | 69.88 (2.92) |
| DFR | ✗ | 60.12 (1.28) | -12.16 (5.34) | -17.36 (3.23) | -27.96 (1.24) | 69.22 (1.25) |
| Sebra | ✗ | 88.61 (3.36) | **-2.21 (3.51)** | -6.89 (3.04) | **-20.36 (2.64)** | **75.36 (2.23)** |

## E METRICS

This section provides a detailed mathematical description of the evaluation metrics used throughout the paper.

1. **In-Domain Accuracy (I.D. Acc)**: This metric represents the weighted average accuracy across groups, where the weights are determined by the correlation strength (i.e., frequency) of each group in the training data. It is designed to assess model performance under conditions where group distribution remains consistent with the training set.

$$\text{I.D. Acc} = \sum_{i=1}^{G} w_i \cdot \text{Acc}_i, \tag{17}$$

where $w_i$ denotes the weight of group $i$, and $\text{Acc}_i$ represents the accuracy for group $i$.

2. **Bias GAP**: This metric captures the difference between In-Domain Accuracy (I.D. Acc) and the accuracy on groups where the specific bias is less pronounced. It quantifies the model's performance drop when tested on groups that diverge from the biases present in the training data.

$$\text{Bias GAP} = \text{I.D. Acc} - \text{Acc}_{\text{uncommon}}, \tag{18}$$

where $\text{Acc}_{\text{uncommon}}$ represents the accuracy on groups with less prevalent bias.

3. **Kendall's Tau Coefficient**: Kendall's Tau is a non-parametric statistic that assesses the ordinal association between two variables. It ranges from -1 (perfect negative correlation) to 1 (perfect positive correlation), with 0 indicating no correlation. Particularly suitable for ranked data, Kendall's Tau is more robust than Pearson's correlation when the data distribution is non-normal or the relationship between variables is non-linear. The coefficient is computed by comparing the number of concordant and discordant pairs in the dataset.

## F   APPLICATIONS BEYOND DEBIASING

This section highlights the wide-ranging applications of Sebra, demonstrating its utility beyond just debiasing tasks. In particular, we demonstrate that sebra faciliates additional utilities beyond debiasing like outlier and noise detections as well as discovery on unknown biases.

### F.1   OUTLIER AND NOISE DETECTION

Training datasets often consist of samples from various sources, annotated by individuals with differing expertise and background knowledge. As a result, it is common for such datasets to contain outliers or mislabeled instances. When these corrupted samples are incorporated into the training process, they can negatively impact model performance, especially if the label noise is prevalent or severe.

Sebra's proposed ranking scheme provides a natural mechanism to address these issues. Specifically, in datasets containing outliers or mislabeled samples, Sebra assigns the highest ranks to these corrupted instances. This ranking system makes it easy to identify and segregate noisy data. For instance, in the Living17 dataset, we demonstrate how Sebra effectively ranks samples across several classes, identifying both the highest- and lowest-ranked samples, as shown in Fig. 8.

The ability to segregate noisy data facilitates an efficient filtration process, which mitigates the negative impact of corrupted samples on model training. This leads to enhanced model robustness and ensures better performance in downstream tasks by focusing on cleaner, more reliable data.

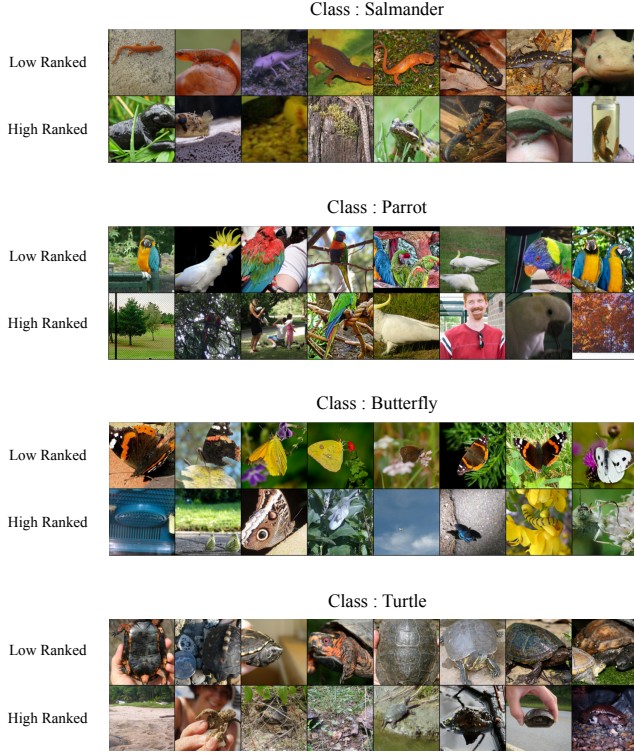

Figure 8: **Qualitative Analysis on the Living17 Dataset.** Examples of the least- and highest-ranked samples from select classes of the Living17 dataset. Sebra assigns higher ranks to mislabeled and outlier samples, enabling their identification and removal in downstream task processing.

### F.2   DISCOVERY OF UNKNOWN BIASES

Datasets often contain various biases, some of which are difficult to identify due to their inherent nature. The Spuriosity ranking generated by Sebra facilitates the discovery of such previously un-

known biases. Through qualitative inspection of the various segregations identified by Sebra, we uncovered two previously unknown biases in widely used synthetic datasets for debiasing studies, as shown in Fig. 9. Specifically, in the UrbanCars dataset, we observed that the color and appearance of the car are spuriously correlated with the 'UrbanCars' category. Similarly, in the Landbirds dataset, a spurious correlation between the color of the birds and the 'Landbirds' category was identified. These subgroups were flagged as relatively high in spuriosity by Sebra, revealing biases that were not previously apparent.

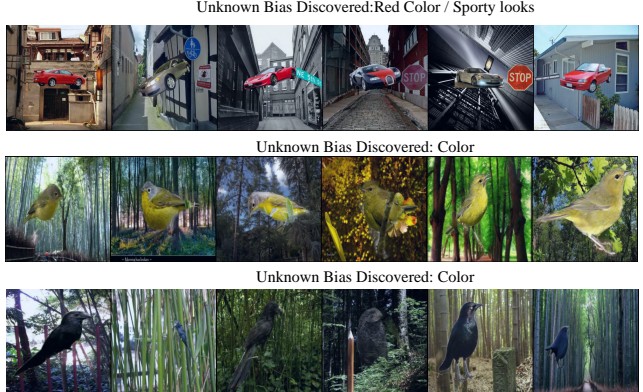

Figure 9: **Discovery of Unknown Biases in Synthetic Datasets.** Examples of spurious correlations uncovered by Sebra in two widely used synthetic datasets for debiasing studies. In the Urban-Cars dataset, a spurious correlation between car color and the 'UrbanCars' category was identified. Similarly, in the Landbirds dataset, a spurious correlation between bird color and the 'Landbirds' category was observed. Sebra's Spuriosity ranking facilitated the identification of these previously unknown biases, aiding in their detection and mitigation.

---

**Algorithm 1:** Pseudocode of Sebra

**Input:** A neural network $f_\theta$, $X_{\text{train}} = \{(x_i, y_i)\}_{i=1}^N$, where $y_i \in \{1, \ldots, C\}$, maximum rank $R$, and upweighting and selection hyperparameters $\beta$ and $\lambda$, respectively.
**Output:** $X_{\text{ranked}} = \{(X_c, \rho(x_i)\}_{c=1}^C$
Initialize $t = 0$
**while** $t < R$ **do**

    Obtain $p_y = f_\theta(x, y)$ to compute $u_i^*$ using equation 3        ▷*Up-weighting*

    Update the model parameters trained with upweighted points using equation 4    ▷*Training*

    Compute $v_i^{t*}$ using equation 2 to select samples for subsequent training    ▷*Selection*

    **if** $v_i^t = 0$ **and** $v_i^{t-1} = 1$ **then**

        $\rho(x_i) = t$        ▷*Ranking*

    Increment $t = t + 1$

---

# G   ADDITIONAL ABLATION STUDIES AND EMPIRICAL VALIDATIONS

## G.1   EFFECT OF VARYING $\beta$

The Sebra objective introduced in Section 3.1 involves two key hyperparameters, $\lambda$ and $\beta$. In this section, we investigate the sensitivity of the proposed ranking scheme to different values of $\beta$. Specifically, we plot the variation of Kendall's $\tau$ metric as a function of increasing $\beta$ values in Fig. 10. As shown, the ranking quality demonstrates an almost linear decreasing trend as $\beta$ increases, suggesting that smaller values of $\beta$ are preferable for optimal performance. This behavior simplifies the hyperparameter search, as the optimal $\beta$ appears to lie within the range $(0, 1)$, reducing the computational cost associated with hyperparameter tuning.

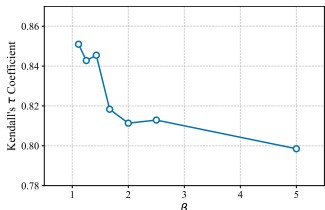

Figure 10: Sensitivity of ranking quality to $\beta$

## G.2 EMPIRICAL VALIDATION OF HARDNESS-SPURIOSITY SYMMETRY

Lin et al. (2022) *et al.*have theoretically demonstrated that unsupervised bias discovery is fundamentally impossible without the incorporation of additional inductive biases or meta-data. In this work, we leverage the concept of *Hardness-Spuriosity Symmetry* as an inductive bias to derive a continuous measure of spuriosity. This symmetry has been explored in prior studies, such as Nam et al. (2020); Qiu et al. (2024). Here, we refine and formalize this concept, proposing a method to quantitatively assess spuriosity.

To empirically validate this assumption, we present a plot of the training loss for samples with and without spurious correlations, generated by training a model using Empirical Risk Minimization (ERM) on the Urbancars, Celeb A and BAR dataset. As shown in Fig. 11, samples containing spurious correlations (i.e., bias attributes) exhibit a rapid decrease in loss, whereas non-spurious samples, which lack such shortcut attributes, show a much slower decline in loss. This discrepancy provides empirical support for our hypothesis that the difficulty of learning from a sample is inversely related to its spuriosity.

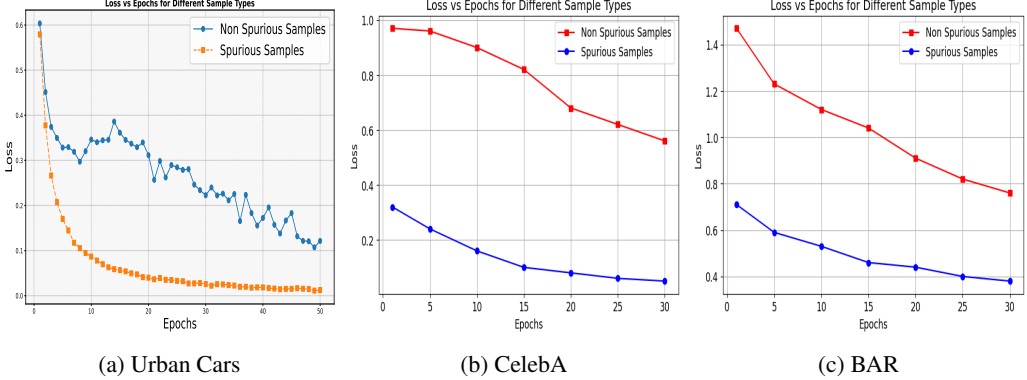

| (a) Urban Cars | (b) CelebA | (c) BAR |

Figure 11: **Empirical Validation of Hardness-Spuriosity Symmetry:** Training loss vs. epochs for samples with and without spurious correlations on the UrbanCars, CelebA and BAR dataset. Samples with spurious correlations demonstrate a rapid decrease in loss compared to samples without such correlations, suggesting that higher spuriosity corresponds to easier learning.

## H COMPUTATIONAL COST

The computational cost of debiasing via self-guided bias ranking can be divided into two components: the cost of spuriosity ranking with Sebra and the cost of contrastive debiasing. Sebra's low computational complexity arises from the closed-form solution for the weighting variable and the progressive removal of data points during ranking, which accelerates the process. However, the cost of bias mitigation using contrastive learning is higher due to its reliance on the full diversity of data rather than a limited subset. To quantify Sebra's computational complexity, we provide a detailed breakdown of the time (in wall-clock minutes) required for both the ranking and debiasing phases of our proposed framework in Table 6. For comparison, we also include the corresponding time for

ERM on the UrbanCars dataset. All the measurements were done using a single Nvidia RTX 3090 GPU.

| Method | Time |
|---|---|
| Sebra - Ranking | 5 minutes |
| Sebra - Contrastive Debiasing (Bias Mitigation) | 52 minutes |
| ERM (Empirical Risk Minimization) | 32 minutes |

Table 6: Time breakdown of Sebra's ranking and bias mitigation steps compared to ERM.

## I  LIMITATIONS

While Sebra demonstrates strong bias ranking capabilities and superior debiasing performance, it remains sensitive to label noise. Another limitation arises in datasets with multiple sub-population shifts, such as class imbalance. In such cases, the model may overemphasize a particular class, resulting in an increasingly unbalanced dataset during training. This imbalance can lead to learning collapse and a failure in ranking performance. Extending Sebra to handle these more complex scenarios, such as bias ranking in the presence of multiple sub-population shifts, could be a promising direction for future research.

## J  REPRODUCIBILITY

In this section, we outline the hyperparameters used in our proposed approach across various datasets. The optimal hyperparameters obtained for various datasets are summarised in Table 7. All experiments were conducted using a single RTX 3090 GPU. To facilitate reproducibility, we intend to release a user-friendly version of the code publicly along with the pre-trained models post-acceptance. We provide all implementation details and hyperparameters to facilitate reproducibility in Table 7. All the datasets used are publicly available or can be generated with publicly available resources.

**Implementation Details:** We use the same architectures and experimental setups as previous studies Li et al. (2022); Nam et al. (2020) to ensure fair comparisons. Specifically, we utilize ResNet-50 for the UrbanCars, and ResNet-18 for CelebA and BAR datasets. The optimal hyperparameters are selected based on experiments conducted on a small validation set with bias annotations, following the approach in Liu et al. (2021); Li et al. (2022) for CelebA and UrbanCars. For BAR, no bias annotations are used, even during validation and validation set is obtained by random split of training set in 80:20 ratio. To ensure statistical robustness, we perform four independent trials with different random seeds and report the mean and standard deviation of the results.

Table 7: Optimal hyper-parameters for the BAR, UrbanCars, CelebA, and ImageNet datasets determined through hyper-parameter search.

| Parameter | UrbanCars | BAR | CelebA | ImageNet |
|---|---|---|---|---|
| Learning Rate (LR) | $1.0 \times 10^{-3}$ | $1.0 \times 10^{-4}$ | $1.0 \times 10^{-3}$ | $1.0 \times 10^{-3}$ |
| batch Size | 128 | 64 | 64 | 512 |
| optimiser | SGD | Adam | SGD | SGD |
| momentum | 0.1 | - | 0.8 | 0.5 |
| weight decay | 0.001 | 0 | $1.0 \times 10^{-4}$ | $1.0 \times 10^{-4}$ |
| $p_{critical}$ | 0.75 | 0.75 | 0.7 | 0.1 |
| $\beta$ | 1.25 | 1.42 | 1.25 | 1.42 |
| $\gamma$ | 0.5 | 0.5 | 1 | 1 |
| $\tau$ | 0.05 | 0.15 | 0.05 | 0.1 |

