# OpenReview forum: "SEBRA : Debiasing through Self-Guided Bias Ranking"
_ICLR.cc/2025/Conference — ICLR 2025 Poster_

### Official Review · Reviewer_APoD · 2024-10-27

**Soundness:** 3
**Presentation:** 3
**Contribution:** 2
**Rating:** 5
**Confidence:** 2

**Summary:**

This work introduces SEBRA (Self-Guided Bias Ranking) to reduce bias in machine learning models without human intervention. Typically, models can develop biases by over-relying on spurious patterns in the data. SEBRA works by identifying these spurious patterns and ranking data points based on how likely they are to contain these misleading cues. During training, SEBRA adjusts the order in which samples are learned, focusing first on those with stronger biases and gradually moving to unbiased samples. This method allows the model to better distinguish between real patterns and biases, resulting in a more accurate and fair model.

SEBRA further incorporates this ranking into a contrastive learning framework, which contrasts data points with strong biases against those with fewer biases, enhancing the model’s ability to learn unbiased representations. Tests across datasets such as UrbanCars, CelebA, and BAR show that SEBRA outperforms other debiasing methods, particularly when multiple sources of bias are present.

**Strengths:**

The paper is easy to understand, especially for readers unfamiliar with the area. The approach is interesting and addresses a problem of importance.

The proposed algorithm seems simple, intuitive yet effective.

The illustration of results in the appendix is quite helpful in building intuition.

**Weaknesses:**

The findings in the SEBRA heavily rely on Assumption 1, termed the Hardness-Spuriosity Symmetry, which states that the difficulty of learning a sample and its spuriosity (presence of misleading features) are inversely correlated. SEBRA uses this assumption to rank and prioritize samples in training, guiding the model to focus on samples with high spuriosity initially. The approach optimizes a ranking objective derived from this assumption, implying that SEBRA's effectiveness in reducing bias is largely contingent on this assumption holding true in practice. If assumption 1 does not hold, SEBRA’s ranking mechanism and bias-mitigation strategy might lose validity, as the model's ranking would no longer accurately reflect the spuriosity of samples.

Please add more references in the related work section.

**Questions:**

Can you delve on the motivation a bit? Why is this kind of de-biasing important?

Is Assumption 1 a part of known literature or is it something discovered during the behavioral exploration of this work? If it is a known fact in debiasing literature, please cite relevant works.

---

> ### Author Response · Authors · 2024-11-21
> **Response to Reviewer APoD**
>
> **W1: The findings in the SEBRA heavily rely on Assumption 1.**
>
> - We agree with the reviewer that SEBRA heavily relies on Assumption 1 for spuriosity ranking. [1] theorically proves that unsupervised bias discovery is fundamentally impossible without further inductive biases or additional information. We adopt 'Hardness-Spuriosness symmetry' as an inductive bias to obtaining a continuous measure of spuriosity. Thus eventhough, Assumption 1 forms a central assumption of the paper, the therorical impossibility result neccesites this.
>
>
>
> **Q1: Can you delve on the motivation a bit? Why is this kind of de-biasing important?**
>
>
> - Training data often contains spurious correlations due to inherent biases or unintended patterns. When deep learning models are trained on such data, they tend to rely on these spurious correlations for predictions. This reliance can lead to poor generalization, particularly when the deployment environment lacks the same spurious patterns present in the training data.
>
> - The consequences of such failures can be severe, especially in critical domains like healthcare, where inaccurate predictions can directly impact patient outcomes. Therefore, it becomes imperative to develop debiasing methodologies that enable models to focus on core, meaningful attributes even when trained on biased datasets. This motivates our proposal to learn a robust, debiased model capable of generalizing effectively from biased data.
>
>
>
> **Q2: Is Assumption 1 a part of known literature or is it something discovered during the behavioral exploration of this work? If it is a known fact in debiasing literature, please cite relevant works.**
>
> - The Hardness-Spuriosity symmetry has been discussed in prior works such as [2, 3]. In this work, we reconcile and formalize this concept to derive a continuous measure of spuriosity. To provide further empirical support for this assumption, we conducted an additional experiment during the rebuttal phase in which we plot the training loss for samples with varying levels of spuriosity in Fig.8 of appendix D4.
>
> - As illustrated in the figure, highly spurious samples are relatively easier to learn, as indicated by the faster decline in their loss compared to samples with low spuriosity.
>
> **References:**
>
> [1] ZIN: When and How to Learn Invariance Without Environment Partition? , NeurIPS 2022
>
> [2] Complexity Matters: Feature Learning in the Presence of Spurious Correlations, ICML 2024
>
> [3] Learning from Failure: De-biasing Classifier from Biased Classifier, NeurIPS 2020

---

> > ### Comment · Reviewer_APoD · 2024-11-23
> >
> > Thank you for the response. I will take a day or two to look into the provided references and then finalize my score.

---

> > > ### Author Response · Authors · 2024-11-30
> > > **Gentle Follow-Up**
> > >
> > > Dear Reviewer APoD,
> > >
> > > As we near the conclusion of the discussion phase, we would greatly appreciate it if you could let us know if you have any remaining questions or concerns. If you find that all issues have been adequately addressed, we would greatly appreciate your consideration in revising the score accordingly. We sincerely hope our rebuttal has effectively responded to all of your feedback.
> > >
> > > Thank you once again for your thoughtful comments and continued engagement.
> > >
> > > Best regards,
> > > The Authors

---

> ### Author Response · Authors · 2024-12-02
>
> Dear Reviewer APoD,
>
> As today marks the conclusion of the discussion phase, we kindly request any additional feedback you may have regarding our submission. We would also appreciate it if you could consider our rebuttal response during your final evaluation.
>
> Thank you for your time and valuable input throughout this process.
>
> Best regards,
>
> Authors

---

> > ### Comment · Reviewer_APoD · 2024-12-03
> >
> > I reviewed the rebuttal revisions (Appendix D4, Fig. 8) and the authors’ response to Reviewer RG64, who raised a similar concern. However, I remain unconvinced that the authors have provided sufficient evidence to substantiate Assumption 1. In my view, it is essential to include theoretical support for this assumption, either in the main paper or in the appendix, as it is foundational to the work. Citing other papers that have used a modified version of this assumption or relying solely on empirical observations from a single dataset is insufficient. I recommend that the authors provide empirical evidence demonstrating that Assumption 1 holds across additional datasets and rewrite this part of the paper. Consequently, I will maintain my current score.

---

> > > ### Author Response · Authors · 2024-12-03
> > > **Additional empirical Support**
> > >
> > > We sincerely thank the reviewer for their valuable feedback. To provide additional empirical support for the assumption, we have extended our analysis to two additional datasets: CelebA and BAR. Below, we present the loss trends for high spurious and low spurious samples across epochs on these datasets.
> > >
> > > ## Results on CelebA
> > >
> > > **Table 1: Loss trends for high and low spurious samples across epochs on the CelebA dataset.**
> > >
> > > | Metric                      | Epoch 1 | Epoch 5 | Epoch 10 | Epoch 15 | Epoch 20 | Epoch 25 | Epoch 30 |
> > > |-----------------------------|---------|---------|----------|----------|----------|----------|----------|
> > > | High Spurious Samples Loss  |  0.32   |  0.24   |   0.16   |   0.10   |   0.08   |   0.06   |   0.05   |
> > > | Low Spurious Samples Loss   |  0.97   |  0.96   |   0.90   |   0.82   |   0.68   |   0.62   |   0.56   |
> > >
> > > ## Results on BAR
> > >
> > > **Table 2: Loss trends for high and low spurious samples across epochs on the BAR dataset.**
> > >
> > > | Metric                      | Epoch 1 | Epoch 5 | Epoch 10 | Epoch 15 | Epoch 20 | Epoch 25 | Epoch 30 |
> > > |-----------------------------|---------|---------|----------|----------|----------|----------|----------|
> > > | High Spurious Samples Loss  |  0.71   |  0.59   |   0.53   |   0.46   |   0.44   |   0.40   |   0.38   |
> > > | Low Spurious Samples Loss   |  1.47   |  1.23   |   1.12   |   1.04   |   0.91   |   0.82   |   0.76   |
> > >
> > > Unfortunately, due to the revision deadline having passed, we are unable to include the corresponding plots in the current version of the paper. However, we will ensure these plots are included in the revised version.
> > >
> > > Additionally, we would like to emphasize that our assumption of *Hardness-Spuriosity symmetry*—i.e., *"the lower the spuriosity, the easier it is to learn"*—is a well-established principle in bias mitigation and has been widely adopted in numerous prior methods[1,2,3,4]. Two very recent developments that provide theoretical evidence for the assumption are [5] and [6], alongside the large body of work on simplicity bias, including but not limited to the prominent results in [7] and [8]. [5] shows that for most practical problems, weights of a DNN converge to the subspace of the input distribution that correspond to the spurious features. More specifically, gradient descent with backpropagation parameterizes the layers with weights that are proportional to the Average Gradient Outer Product (AGOP) of the input features with respect to the function computed by the corresponding layers. The AGOP is essentially the subspace of the input that is maximally sensitive to perturbations - i.e., for which a small change in the input leads to a large change in the output. Combining this with the already well-known simplicity bias of neural networks [7, 8], the authors of [5] showed that the less spurious a feature is, the harder it is to learn for a neural network, which is essentially the same as our Hardness-Spuriosity Symmetry assumption.
> > >
> > > [6] delved deeper into the mechanics of representation learning to investigate the relationship between the simplicity bias, the depth of a neural network, and its proclivity to spurious correlations. Based on the result that deeper networks learn features that lie on a lower rank manifold [8], the authors of [6] showed that deeper networks are also more vulnerable to be dependent on spurious features, since the spuriosity of a feature is inversely correlated with its rank. In other words, as the spuriosity of a feature increases, its rank goes down, which consequently implies that it can be encoded with fewer dimensions, and is hence, easier to learn. This work again theoretically supports the validity of our Hardness-Spuriosity Symmetry assumption. We will include these discussions on the theoretical evidence for our assumption in the final version of our paper.
> > >
> > > We hope that the experimental results we have provided on the two additional datasets, along with the discussion on the theoretical evidence for our assumption, address your concerns, and we kindly request that you take them into account in your evaluation.
> > >
> > >
> > > References:
> > >
> > > [1] Complexity Matters: Feature Learning in the Presence of Spurious Correlations, ICML 2024
> > >
> > > [2] Learning from Failure: De-biasing Classifier from Biased Classifier, NeurIPS 2020
> > >
> > > [3] Avoiding spurious correlations via logit correction." ICLR, 2023.
> > >
> > > [4] Training Debiased Subnetworks with Contrastive Weight Pruning, CVPR 2023
> > >
> > > [5] Radhakrishnan et al., "Mechanism for feature learning in neural networks and backpropagation-free machine learning models", Science 2024.
> > >
> > > [6] Sreelatha et al., "DeNetDM: Debiasing by Network Depth Modulation", NeurIPS 2024.
> > >
> > > [7] K. Hermann and A. Lampinen, "What shapes feature representations? Exploring datasets, architectures, and training", NeurIPS, 2020.
> > >
> > > [8] Huh et al., "The low-rank simplicity bias in deep networks", TMLR 2023.

---

### Official Review · Reviewer_RG64 · 2024-10-28

**Soundness:** 2
**Presentation:** 3
**Contribution:** 2
**Rating:** 6
**Confidence:** 3

**Summary:**

This paper proposes a bias mitigation method by ranking data points based on the assumption that the harder a data point is to learn, the less spuriosity it has and the overall ranking process is achieved by selecting and upweighting the samples that have not yet been learned, which relive the need of human supervision compared to existing method used to identify biased features.

**Strengths:**

- The paper addresses the bias mitigation problem by introducing a fine-grained estimation method instead of the traditional binary biased-*vs*-unbiased partitioning approach, providing a more precise way to evaluate spuriosity and the combination of the contrastive learning framework is a nice attempt to further reduce the reliance on the few unbiased samples compared to existing methods by utilizing all the samples in the dataset.
- The intuition behind the proposed method is clearly with the definitions and theorems. Although I think the assumption (assumption 1, Hardness-Spuriosity Symmetry) is strong, it still demonstrates a clear logic explaining why the ranking mechanism is designed in this way.

**Weaknesses:**

- The proposed method heavily relies on the assumption 1 (Hardness-Spuriosity Symmetry), which states that samples with large losses (difficult to learn) are likely to be less spurious. This raises concerns about the method’s effectiveness when the samples are outliers or have label noise issues....so if the large loss is caused by these samples, then utilizing them for training would adversely affect the performance on the downstream tasks. I think such condition should be discussed.
- The proposed method has three parameters to optimize, so when the dataset has high dimensionality, the optimization process may become very slow, and potential convergence issues should be discussed.
- The overall writing of this paper is not very straightforward, as it repeats the main concepts too many times (e.g., the concept of spuriousness, the assumption, selection, upweighting.....) and the method section (3.1 to 3.3) has very weak connections, making each part appear independent of the others, which weakens the logical flow when introducing the method. I suggest the authors consider merging some of the repeated content (or summarize when it repeat) and adding some transitional words between sections to improve coherence.

**Questions:**

- When selecting points that have not yet been learned, which are defined by large loss, how can the method avoid selecting data points that are outliers or affected by label noise issues?

- Is Assumption 1 (Hardness-Spuriousness Symmetry) supported by any literature or empirical evidence?

---

> ### Author Response · Authors · 2024-11-21
> **Response to Reviewer RG64**
>
> We thank the reviewer for their constructive feedback and thoughtful suggestions. Below, we provide detailed clarifications addressing the questions and concerns raised.
>
> **W2: Optimization Parameters**
>
> - The method involves three parameters that influence the optimization process. However, two of these parameters, $v_i$ and $u_i$, have closed-form solutions, as presented in Equations 2 and 3 of the paper. Since these solutions do not require optimization, the number of parameters that need to be optimized is reduced to just one.
>
> - Additionally, the convergence time of Sebra is primarily influenced by the size of the training dataset rather than the dimensionality of the data. This makes Sebra scalable to high-dimensional datasets. Furthermore, the ranking phase of Sebra is very efficient, taking only 5 minutes on the UrbanCars dataset, which indicates quick convergence.
>
>
> **W1/Q1: Handling Outliers and Label Noise**
>
> - We appreciate the reviewer’s concern regarding the potential failure of **Sebra** in datasets with outliers or label noise. Sebra ranks data points within each class based on their spuriousness. In the presence of label noise or outliers, such samples tend to be ranked highest in the obtained rank order. Since Sebra isolates these outliers and mislabeled samples in the highest-ranked set, they can be excluded from the contrastive debiasing stage. This exclusion effectively mitigates the negative influence of such samples on the debiasing process.
>
> - To provide empirical evidence, we applied our ranking scheme to the Living17 dataset, which contains 17 classes derived from ImageNet. Qualitative visualizations of the highest- and lowest-ranked samples for several classes are provided in Fig 7, Appendix D3. As shown in the figure, the highest-ranked images (bottom row) consist predominantly of outliers and mislabeled samples, while the lowest-ranked images (top row) are devoid of such issues. By discarding the highest-ranked samples, the method can effectively circumvent the adverse effects caused by outliers or noisy labels.
>
>
>
> **Q2: Support for Assumption 1 (Hardness-Spuriousness Symmetry)**
>
> - The Hardness-Spuriosness symmetry has been discussed in some of the prior works such as [1,2],  which we reconcile and formalize for the purpose of obtaining a continuous measure of spuriosity. Sebra distinguishes itself from these works by going beyond a simple binary partitioning of the training data. Instead, it produces a per-class rank ordering of the data. This additional layer of derived knowledge enables more efficient utilization of the diversity within the training data, leading to superior bias mitigation. To provide further empirical support for 'Hardness-Spuriousness Symmetry', we provide a plot of training loss of samples with different levels of spuriosity in Fig 8. of appendix D.3.
> As illustrated in Fig.8 highly spurious samples are relatively easy to learn indicated by a faster decline in their loss compared to samples with low spuriosity.
>
> References:
>
> [1] Complexity Matters: Feature Learning in the Presence of Spurious Correlations, ICML 2024
>
> [2] Learning from Failure: De-biasing Classifier from Biased Classifier, NeurIPS 2020

---

> > ### Comment · Reviewer_RG64 · 2024-11-22
> > **All concerns addressed, revised version improves clarity**
> >
> > Thanks to the authors for providing detailed responses and the effort the authors have put into addressing my questions and improving the manuscript for greater clarity.
> > - Regarding the optimization parameters, my concerns have been addressed.
> > - Regarding the outliers and label noise, I appreciate the added discussion in Appendix D.3 and the clarification provided. I find Figure 7 to be a good illustration of how SEBRA identifies outliers and samples with label noise.
> > - Regarding Assumption 1 (which was my main concern), I noticed that Reviewer APoD shares a similar concern. While the empirical validation provided in Appendix D.4 (Figure 8) is helpful, I believe it might be better to highlight this point in the main text (perhaps a brief mention in the introduction, noting that this assumption is supported by the empirical evidence in Appendix D.4), could strengthen the manuscript further, since this is a crucial foundation that supports the SEBRA algorithm.
> >
> > Overall, since all my concerns and questions have been addressed and clarified, and I have reviewed the revised version, which incorporates my suggestions for improving conciseness and coherence, I am happy to raise my score.

---

> > > ### Author Response · Authors · 2024-11-22
> > > **Thank you!**
> > >
> > > Dear Reviewer RG64,
> > >
> > > We extend our gratitude for raising our score. We're sincerely thankful for your support! We will include all the updates in our revision.
> > >
> > > Best regards,
> > >
> > > The Authors

---

> ### Author Response · Authors · 2024-11-22
> **Overview of Writing Revisions**
>
> **W3: The overall writing of this paper is not very straightforward:** We thank the reviewer for their suggestions in improving the writing of our paper in terms of conciseness and coherence. Based on their recommendations, we have now incorporated several updates across the Methodology and the Introduction sections, which are as follows:
> - Merged several redundant portions of text such as the discussions on the concept of spuriousness, the assumption of the Hardness Spuriosity Symmetry, descriptions of the various phases of Sebra such as Selection, Upweighting, and Ranking, etc.
> - Added connecting sentences around the Methodology subsection (3.1 to 3.3) to improve coherence.
> - Substituted some of the abstract descriptions of our methodology with an intuitive example in Section 3.1, Intuition behind Sebra.

---

> ### Author Response · Authors · 2024-11-22
> **Summary of Additional Experiments**
>
> We summarise the additional set of experiments that were performed during the rebuttal phase:
>
> - Additional qualitative results on the Living 17 dataset (Appendix D.3 and Fig.7), to demonstrate the capability of Sebra in segregating and isolating outliers and mislabeled samples.
>
> - Additional empirical evidence supporting hardness-spuriosity symmetry (Appendix D.4 and Fig.8)

---

### Official Review · Reviewer_CwjJ · 2024-11-01

**Soundness:** 3
**Presentation:** 3
**Contribution:** 2
**Rating:** 6
**Confidence:** 3

**Summary:**

The authors mainly aim to address spurious correlations by ranking instances according to the extent of biases automatically and conducting contrastive learning with these rankings. Specifically, based on the observation that the extent of biases is proportional to learning speed, they modify the learning process of ERM to progressively capture biases according to their extent. This modification process consists of three phases: selection, upweighting & training, and ranking. Then, with acquired rankings, they conduct contrastive learning to debias models by constructing negative pairs with similar levels of spurious correlations and positive pairs with different levels of spurious correlations. In experiments, the authors demonstrate the effectiveness of their method.

**Strengths:**

* The proposed method can acquire the ranking of instances based on the extent of spurious correlation without human annotations.
* The authors show that their ranking is similar with the ground truth ranking and the overall framework mitigates performance gaps.
* The paper is well-written.

**Weaknesses:**

- The paper lacks a detailed discussion on the utility of ranking instances according to the strength of spurious correlation. In this paper, the ranking is used for debiasing models via contrastive learning, yet it remains unclear how this approach offers high-level advantages over debiasing methods that do not utilize ranking. It would enhance the paper to discuss potential applications of their fine-grained spuriosity rankings beyond the contrastive learning framework. For example, the recent method, B2T [1], can identify and mitigate biases without human supervision by leveraging keywords. Since keywords are natural language, humans can interpret identified biases as well. Would the authors clarify the advantages over B2T?
- The experimental validation of the proposed method is not convincing. First, the comparison does not include recent debiasing methods [1, 2, 3]. Additionally, the model is validated only on a few datasets within the visual domain. The paper would be strengthened by including experiments on other widely-used datasets for spurious correlation, such as Waterbird, and NLP tasks like MultiNLI and CivilComments. Finally, the metrics are not commonly used in the spurious correlation domain, so reporting worst-group accuracy would enhance comparability.

[1] Kim, Younghyun, et al. "Discovering and Mitigating Visual Biases through Keyword Explanation." CVPR, 2024.

[2] Liu, Sheng, et al. "Avoiding spurious correlations via logit correction." ICLR, 2023.

[3] Deng, Yihe, et al. "Robust learning with progressive data expansion against spurious correlation." NeurIPS, 2023.

**Questions:**

* Would the authors provide how much additional training time is required compared to training ERM. It would be helpful if you could provide a breakdown of the time spent on ranking and contrastive learning separately. Although there is a discussion in the appendix, specific values are not provided.

---

> ### Author Response · Authors · 2024-11-21
> **Response to Reviewer CwjJ[1/3]**
>
> We appreciate the reviewer’s insightful suggestions and comments. Below, we provide detailed responses to the questions and concerns raised.
>
>
>
> **W1-[1/3]: How does this approach offer high-level advantages over debiasing methods that do not utilize ranking?**
>
> - Most prior debiasing methods primarily partition the training data into two subsets: bias-aligned(samples with spurious correlations) and bias-conflicting samples(samples without spurious correlations). These methods predominantly rely on identified bias-conflicting samples for bias mitigation. However, the number of bias-conflicting data points is typically very small, often comprising only 0.5% or even less, particularly for datasets with multiple biases. This limited reliance on a small subset of samples prevents these methods from fully leveraging the diversity of the bias-aligned samples in the training set. We have discussed this briefly in section 3.3 (lines 340-345), and we will revise this for additional clarity.
>
> - In contrast, Sebra's ranking framework enables a more effective utilization of the diversity inherent in bias-aligned samples. By leveraging the obtained rankings, Sebra facilitates the construction of positive and negative pairs even within the bias-aligned subset. This approach allows the model to learn target attributes from a broader range of samples, including bias-aligned data, thereby harnessing the full diversity of the training dataset. As a result, Sebra does not require complex data augmentation strategies like many of the recent debiasing works like Logit Correction.
>
> - Furthermore, the framework of spuriosity ranking was introduced in [1] though they rely on human supervision. Sebra enables the same without reliance on human supervision.

---

> ### Author Response · Authors · 2024-11-21
> **Response to Reviewer CwjJ[2/3]**
>
> **W1-[2/3]: Potential applications of fine-grained spuriosity rankings beyond the contrastive learning framework?**
>
> - The **contrastive debiasing framework** proposed in our method is a design choice adopted to enable efficient debiasing. Beyond the contrastive learning framework, fine-grained spuriosity ranking could be combined with other approaches, such as **DRO (Distributionally Robust Optimization)**, which involves the minimization of worst-group loss. The groups in this scenario could be formed by utilizing the ranks obtained by Sebra. Other alternatives could utilize last-layer retraining-based approaches or leverage metric learning to enable efficient debiasing.
>
> - Beyond debiasing, the fine-grained spuriosity ranking could also be utilized for:
>
>     - **Revealing minority subpopulations**: High-rank images obtained via Sebra's rankings often correspond to minority subpopulations in the data where spurious correlations are absent.
>
>     - **Mislabelled samples and Outlier Identification**:  Sebra's rankings can assist in identifying mislabeled samples. For example, high-ranked images within each class are more likely to include mislabeled instances. We present qualitative results in Appendix D3 (Figure 7), providing empirical evidence to support this claim.

---

> ### Author Response · Authors · 2024-11-21
> **Response to Reviewer CwjJ[3/3]**
>
> **W1-[3/3]: Advantages over B2T**
>
> - We thank the reviewer for drawing attention to this closely related work. While B2T demonstrates strong debiasing performance, it faces key limitations when compared to Sebra:
>
>     - **Inherited spurious correlations from pre-trained models**:
>        B2T relies on pre-trained CLIP models for bias identification. However, vision-language models like CLIP are known to contain various biases [2]. If there is an overlap between biases in the dataset and those captured by the pre-trained model, the bias identification process may be compromised. In contrast, SEBRA does not depend on any pre-trained models that might contain spurious correlations, making it more robust in such scenarios.
>
>     - **Higher Computational Cost**: B2T incur significant computational overhead due to the need for caption generation and CLIP score computation. This overhead can increase considerably depending on the complexity of the model used. In contrast, Sebra leverages smaller models, resulting in improved computational efficiency for bias discovery compared to B2T.
>
>     - **Limitations on non-natural images**:
>        B2T relies heavily on the rich knowledge acquired through pretraining on large-scale natural image datasets. However, such pre-trained models may not generalize to datasets from other domains (e.g., medical or satellite imagery). As demonstrated in Appendix F of the B2T paper, pre-trained CLIP models fail on datasets outside their natural image domain. Addressing this limitation often requires pre-trained vision-language models aligned with the specific dataset, which can be challenging to obtain. SEBRA, by design, does not rely on such pre-trained models, making it broadly applicable across diverse domains.
>
>
> **W2.1: Reporting Worst-Group Accuracy (WGA)**
>
> - We agree with the reviewer that reporting WGA enhances comparability. Below, we provide the Worst-Group Accuracy (WGA) for datasets with group information:
>
> |           | UrbanCars   | CelebA     |
> |-----------|-------------|------------|
> | ERM       | 33.2 (0.86) | 36.0 (1.7) |
> | Group-Dro | 75.7 (1.79) | 37.9 (1.6) |
> | LfF       | 35.6 (2.40) | 35.5 (2.0) |
> | JTT       | 33.3 (6.9)  | 38.7 (2.4) |
> | Debian    | 30.1 (0.89) | 41.1 (4.3) |
> | Sebra     | 73.8 (3.28) | 65.3 (4.1) |
>
>
> As shown in the table below, Sebra demonstrates competitive WGA, significantly outperforming other methods like ERM, LfF, and JTT, particularly excelling on CelebA (65.3) and achieving near Group-Dro levels on UrbanCars (73.8). These results will be added to our revised submission.
>
>
> **W2.2: Additional Datasets and Baselines**
>
> - We are actively extending our baselines and datasets to ensure broader coverage and comparability. We aim to share these results before the end of the discussion phase.
>
>
>
> **Q1: Computational Complexity**
>
> - To better illustrate **Sebra's** computational complexity, we provide a detailed breakdown of the time (in wall-clock minutes) required for both the ranking and debiasing phases of our proposed framework. For comparison, we also include the corresponding time for **ERM** on the UrbanCars dataset. All the measurements were done using a single Nvidia RTX 3090 GPU.
>
>     ### Time Comparisons
>     | **Method**                                      | **Time**        |
>     |-------------------------------------------------|-----------------|
>     | Sebra - Ranking                                 | 5 minutes   |
>     | Sebra - Contrastive Debiasing (Bias Mitigation) | 52 minutes  |
>     | ERM (Empirical Risk Minimization)               | 32 minutes  |
>
>
>
>
> - Please let us know if further clarifications or additional experiments would strengthen our response. We greatly appreciate the reviewer’s time and thoughtful feedback.
>
> **References:**
>
> [1]Spuriosity Rankings: Sorting Data to Measure and Mitigate Biases, NeurIPS 2023
>
> [2] CLIP the Bias: How Useful is Balancing Data in Multi modal Learning? ICLR 2024

---

> ### Comment · Reviewer_CwjJ · 2024-11-23
> **Remaining concerns**
>
> I thank the authors for their detailed replies to my questions and concerns.
>
> Most of my concerns and questions have been addressed, but the effectiveness of the proposed method remains insufficient. Methods like LfF, JTT, and SelecMix also utilize bias-aligned samples in training, and it is not yet fully convincing whether the proposed approach has a clear advantage over these methods at a high level. Therefore, it is crucial to demonstrate experimentally that the proposed method leverages bias-aligned samples more effectively, outperforming other methods. However, this aspect appears to be inadequately addressed.
>
> I appreciate the additional experiments provided, but I noticed that the baseline performance on CelebA differs significantly from the performance reported in JTT. For a fair comparison, it is important to present results aligned with the JTT experimental setup. Furthermore, the authors have not yet shared results demonstrating the proposed method’s strengths in NLP tasks. I will finalize my score after reviewing these results.

---

> > ### Author Response · Authors · 2024-11-26
> > **Clarification on remaining concerns [2/2]**
> >
> > **Q2.Why does the baseline performance on CelebA differ significantly from the performance reported in JTT?**
> >
> > The performance discrepancy between the results reported in JTT paper and those in our submission arises from the fact that we used a more challenging variant of the CelebA dataset, which includes multiple biases, as introduced in [1] (discussed on lines 382-383 of the main text.). In contrast, the results reported in JTT are based on a single-bias setting.
> >
> > ---
> >
> > **Q3:Effectiveness of Sebra on NLP datasets?**
> >
> >
> > We are actively investigating the applicability of Sebra to NLP datasets and are confident about obtaining insightful results soon. In the worst case, even if Sebra encounters challenges in delivering strong performance on NLP tasks in the current format, we will accordingly narrow the scope of our work down to focus on addressing visual biases, a well-established and significant research problem, as highlighted in [2,3]. In such a case, generalizing Sebra to NLP tasks would be a promising and exciting direction for future exploration. So we consider this NLP evaluation would be optional without affecting our major contributions. But we definitely appreciate this insightful comment.
> >
> > ---
> >
> > ## References:
> > 1. Rui Hu et al., "Unsupervised Debiasing via Pseudo-Bias Labeling in an Echo Chamber."
> > 2. Younghyun Kim et al., "Discovering and Mitigating Visual Biases through Keyword Explanation."
> > 3. Jungsoo Lee et al., "Learning Debiased Representation via Disentangled Feature Augmentation"

---

> > > ### Author Response · Authors · 2024-11-30
> > > **Results on NLP Tasks**
> > >
> > > We sincerely hope that our rebuttal has addressed all of your remaining concerns. In addition, we conducted an experiment with our framework on one of the NLP tasks you suggested, specifically **MultiNLI**. The results of this experiment are summarized in the table below:
> > >
> > > | **Method**             | **Worst Group Accuracy** |
> > > |-------------------------|--------------------------|
> > > | ERM                    | 66.8                     |
> > > | GroupDRO (Supervised)  | 76.0                     |
> > > | LfF                    | 63.6                     |
> > > | JTT                    | 69.1                     |
> > > | Sebra                  | 72.3                     |
> > >
> > > As illustrated above, Sebra demonstrates higher worst-group accuracy compared to all unsupervised baselines, further validating its effectiveness and applicability to NLP tasks. It is important to note that while GroupDRO is a supervised method, Sebra is an unsupervised approach, and we outperform all unsupervised SOTA models. The baseline results are taken from [1].
> > >
> > > As the discussion period is coming to an end, we would be truly grateful if you could take a moment to review our response to your comments. Should you have any further questions or require clarification, please don’t hesitate to let us know. If everything is clear, we would sincerely appreciate it if you could consider revising the score in light of our updates. We look forward to your feedback and are happy to address any remaining concerns before the discussion period ends.
> > >
> > > #### References
> > > [1] *Change is Hard: A Closer Look at Subpopulation Shift*, ICML 2023

---

> ### Author Response · Authors · 2024-11-26
> **Clarification on remaining concerns [1/2]**
>
> We sincerely thank the reviewer for their active engagement during the discussion phase and are pleased to note that most of the concerns have been addressed. Below, we provide further clarifications on the remaining points:
>
> **Q1. Whether the proposed approach have a clear advantage over other methods at a high level?**
>
> Methods like **LfF** and **JTT** explicitly downweight the bias-aligned samples during debiased training relative to bias-conflicting samples, thereby restricting the model's learning from bias-aligned samples:
>
> - **JTT** upweights the contribution of bias-conflicting points by a factor $\lambda_{up}$ (typically taking values between 50–100). As a result of this relative upweighting, the contribution of bias-aligned points is reduced.
> - **LfF** upweights training samples using a derived weighting factor $W(x)$, with biased samples receiving smaller weights, again reducing their overall contribution to debiased learning.
> - **SelectMix** partitions the training data into bias-aligned and bias-conflicting sets, then utilizes MixUp for augmenting the bias-conflicting samples. Even though there is no explicit down-weighting of bias-aligned samples in this case, all the bias-aligned samples are treated alike (there is no distinction between severely bias-aligned and slightly bias-aligned points). As a result:
>   - During the MixUp phase, slightly bias-conflicting samples might be augmented using severely bias-aligned samples, making the resulting sample predominantly biased. These points, when utilized for training, could cause the model to capture undesirable bias attributes.
>
> ### Advantages of Sebra
>
> **Sebra** offers clear advantages over these previous methods in the following aspects:
>
> - Unlike LfF and JTT, Sebra does not downweight any of the training points, enabling the model to learn from all the data points. This is possible because the fine-grained ranking obtained allows the formation of informative contrastive pairs, enabling the model to learn core attributes even from bias-aligned points.
>
> - Unlike in select mix, wherein often the augmented samples generated by mixup could be highly biased (as described above), our approach because of the availability of the ranking would only contrast a highly bias-aligned point with highly bias-conflicting samples.
>
> We would like to emphasize additional key advantages that set our framework apart from prior debiasing approaches:
>
> - **Mitigating Multiple Biases:** Unlike previous methods (including LfF, JTT, and SelectMix), which struggle when datasets contain multiple biases, Sebra can mitigate multiple biases simultaneously.
>
> - **Applications Beyond Debiasing:** Sebra goes beyond traditional debiasing by enabling valuable applications such as the identification of mislabeled samples and outliers, broadening its utility across various tasks.
>
>
> ### Experiemental Support
>
> We deeply appreciate the reviewer for highlighting the lack of evidence supporting our claim that Sebra leverages bias-aligned points more efficiently compared to existing methods. This is a critical observation, and we recognize its importance in establishing the credibility of our results.
>
> To address this gap, we conducted an additional experiment on the **BAR dataset**. In this experiment, we selected a couple of the state-of-the-art (SOTA) methods identified by the reviewer (JTT, LfF) and evaluated their performance after excluding the bias-aligned points from the training set. We followed the same procedure as our proposed method, Sebra. Specifically:
>
> | Method                     | Accuracy (%) | $\Delta$   |
> |----------------------------|--------------|------------|
> | **JTT (without bias-aligned)** | 67.24        | —          |
> | **JTT**                    | 68.14      | +0.9      |
> | **LfF (without bias-aligned)** | 66.4            | —          |
> | **LfF**                    |  68.30    | +1.9          |
> | **Sebra (without bias-aligned)** | 68.0         | —          |
> | **Sebra**                  | 75.36        | +7.36      |
>
> Here, **$\Delta$** denotes the improvement in test accuracy upon the introduction of bias-aligned training samples.
>
> As observed in the above table, the inclusion of bias-aligned points to the training paradigm of JTT, and LfF only marginally impacted the performance by 0.9% and 1.9% respectively. Debiasing using Sebra produces a significant improvement of 7.36% supporting our claim that Sebra leverages bias-aligned points more effectively compared to existing methods.

---

> ### Comment · Reviewer_CwjJ · 2024-12-01
>
> I appreciate the detailed response. Most of my concerns are addressed and I will raise my score.

---

> ### Author Response · Authors · 2024-12-01
> **Thank you**
>
> Dear Reviewer CwjJ,
>
> Thank you for your constructive comments and for raising your score! We greatly appreciate your support!
>
> Best regards,
>
> The Authors

---

### Official Review · Reviewer_GZFw · 2024-11-04

**Soundness:** 3
**Presentation:** 2
**Contribution:** 3
**Rating:** 6
**Confidence:** 2

**Summary:**

This paper aims to rank samples by the degree of spuriosity and debias accordingly. The authors propose an unsupervised framework, Sebra, which ranks samples automatically by spuriosity in their classes. An important assumption of this paper is the negative correlation between spuriosity and the ease of learning a sample via ERM. Further, they dynamically steer ERM to satisfy this assumption during iteration training and sequentially learn samples in increasing order of difficulty. Experiment results show superiority of Sebra over other unsupervised debiasing methods.

**Strengths:**

- The motivation of this paper is simple and clear: biased samples are harder to learn.
- The design of methods is well described and easy to understand.
- Performance improvement is significant.

**Weaknesses:**

- The method parts are a little too long and not eye-catching, written in a way that follows the authors' thought processes. I would prefer it to be more concise and add more examples or illustration figures.

- Some sentences are way too long to understand, e.g., lines 340-343. Also, plz pay attention to distinguishing \cite and \citep.

- Class labels are required in this method.

**Questions:**

- Can this method be extended to semi-supervised, especially when encountering new classes?

---

> ### Author Response · Authors · 2024-11-21
> **Response to Reviewer GZFw**
>
> We thank the reviewer for their valuable suggestions. Below, we address your concerns in detail:
>
> **Q1: Extension to Semi-Supervised Learning with New Classes**
>
> - Sebra is orthogonal conceptually to the learning paradigms (e.g., supervised, semi-supervised learning), as no added assumption is made on the amount of labels and if new classes present or not. Semi-supervised learning is not evaluated in our initial submission since supervised learning is the setting adopted in all existing works, where debiasing remains a challenge. Extending to more challenging setups like the suggested is critical as part of future work. One potential approach is to initialize the backbone architecture with weights obtained from unsupervised pretraining on unlabeled data, and then apply Sebra to achieve rank ordering on labeled data. For instance, in the case of the UrbanCars dataset, our training protocol follows a similar strategy: we first initialize the feature extractor with pretrained ImageNet weights and then proceed to obtain a rank order for the two classes of UrbanCars (urban/country), which are not included in the ImageNet dataset. We agree that extending our framework to the semi-supervised setting is quite interesting, but our paper focuses on the bias in supervised learning and makes a unique and important contribution to this research area by proposing the ranking-based debiasing method.
>
> We plan to provide additional experimental evidence to further demonstrate the applicability of Sebra in a semi-supervised learning framework with new classes before the conclusion of the rebuttal period.
>
>
> **W1, W2: Improvement in Writing**
>
> - We appreciate the reviewer’s feedback and suggestions for improving the writing, as well as for pointing out some overlooked errors. We would incorporate these suggestions in further revisions. Additionally, we will move some of the qualitative results from the appendix to the main paper to provide more examples and improve the overall clarity.
>
> **W3: Requirement of Class Labels**
>
> - We appreciate your observation regarding the use of class labels. To clarify, our method requires labels for the target variable as it operates within a supervised learning framework, consistent with prior works in this domain. In the given problem setting, spurious correlations are defined and measured with respect to the target variable, making target labels an inherent part of the problem setting. Our method does not introduce any additional label requirements beyond those specified by the problem setting.
>
> - Importantly, our approach does not require labels for the protected attribute, a key aspect of its design that ensures applicability in scenarios where such information is unavailable or unreliable.

---

> > ### Comment · Reviewer_GZFw · 2024-11-22
> >
> > Thank you for your responses, I will keep my positive score.

---

> > > ### Author Response · Authors · 2024-11-22
> > > **Thank you!**
> > >
> > > Dear Reviewer GZFw,
> > >
> > > Thank you for your valuable suggestions and for retaining the positive score! We will incorporate all the changes in our revised submission.
> > >
> > > Best regards,
> > > The Authors

---

### Meta-Review · Area_Chair_sFAQ · 2024-12-19

**Metareview:**

This paper introduces a new bias ranking method that organizes data in descending order of spuriosity without requiring human supervision. Initially, the paper received mixed reviews before the rebuttal. However, following a constructive rebuttal, most concerns are successfully addressed. The only remaining major issue is raised by the reviewer 'APoD', who questions the validity of the assumption. After reviewing the paper and discussions, I believe this concern is not a big issue for the following reasons:

- The authors have conducted additional experiments in their latest response, supporting the assumption.
- Similar assumptions have been utilized in other studies.

Overall, I believe the advantages of this paper outweigh its disadvantages. I recommend accepting it, with the authors required to include all rebuttal details in the final version.

**Additional Comments On Reviewer Discussion:**

After rebuttal, all the reviewers except `APoD` have a positive score.  The only remaining major issue is raised by the reviewer 'APoD', who questions the validity of the assumption. After reviewing the paper and discussions, I believe this concern is not a big issue for the following reasons:

- The authors have conducted additional experiments in their latest response, supporting the assumption.
- Similar assumptions have been utilized in other studies.

Overall, I believe the advantages of this paper outweigh its disadvantages. I recommend accepting it, with the authors required to include all rebuttal details in the final version.

---

### Decision · Program_Chairs · 2025-01-22

Accept (Poster)